



# Operational wind plants increase planetary boundary layer height: An observational study

Aliza Abraham[1], Matteo Puccioni[2], Arianna Jordan[3], Emina Maric[1], Nicola Bodini[1], Nicholas Hamilton[1], Stefano Letizia[1], Petra M. Klein[3], Elizabeth Smith[4], Sonia Wharton[2], Jonathan Gero[5], Jamey D. Jacob[6], Raghavendra Krishnamurthy[7], Rob K. Newsom[7], Mikhail Pekour[7], and Patrick Moriarty[1]

[1]National Renewable Energy Laboratory, Golden, Colorado, USA
[2]Lawrence Livermore National Laboratory, Livermore, California, USA
[3]University of Oklahoma School of Meteorology, Norman, Oklahoma, USA
[4]NOAA/OAR/National Severe Storms Laboratory, Norman, Oklahoma, USA
[5]University of Wisconsin Space Science Engineering Center, Madison, Wisconsin, USA
[6]Oklahoma State University School of Mechanical and Aerospace Engineering, Stillwater, Oklahoma, USA
[7]Pacific Northwest National Laboratory, Richland, Washington, USA

**Correspondence:** Aliza Abraham (aliza.abraham@nrel.gov)

**Abstract.** As wind energy deployment grows, interactions between wind plants and the surrounding environment become more prevalent. The current investigation seeks to understand these interactions by characterizing the impact of wind plants on the planetary boundary layer height (PBLH), utilizing observations from the American WAKE ExperimeNt (AWAKEN) campaign. Given the ambiguity of the definition of PBLH under stable atmospheric conditions, where the impact of wind plants

is expected to be strongest, a comparison of different methods for identifying PBLH is first conducted using data collected by multiple different instruments. Then, using one of these methods that is thermodynamic and another that is turbulence-based, the values of PBLH measured at spatially distributed sites are compared under a range of atmospheric conditions. Both methods show a clear increase in PBLH downstream of a wind plant for stable conditions. These impacts are strongest when the upstream PBLH is shallow (less than 0.25 km), with the thermodynamic method showing a PBLH increase of 33–39 %

and the turbulence-based method showing a 141 % increase. At a site 20 km downstream of the wind plant, these effects are no longer observed, suggesting PBLH has recovered. The results of this investigation show that wind plants can modify the surrounding atmosphere, improving understanding of wind plant–atmosphere interaction that is crucial for model development and validation.

## 1 Introduction

Wind energy deployment continues to increase, necessitating an improved understanding of the interactions between wind plants and their surroundings. Turbines extract energy from the wind, which reduces the speed of the flow and modifies the turbulence in their wakes (Vermeer et al., 2003; Bodini et al., 2021). These wake effects can also induce changes in atmospheric properties such as surface temperature and momentum fluxes (Zhang et al., 2013; Abraham and Hong, 2021). When tens to hundreds of turbines are arranged into a wind plant, their combined wakes have been shown to influence air temperature,



humidity, and vegetation (Baidya Roy et al., 2004; Baidya Roy and Traiteur, 2010; Lu and Porté-Agel, 2011; Zhou et al., 2012; Smith et al., 2013; Armstrong et al., 2016; Li et al., 2018a; Miller and Keith, 2018; Xia et al., 2019). The magnitude of these wind turbine and plant wake effects vary with atmospheric conditions, and many studies have shown that they are stronger in a stable atmosphere, where ambient turbulence is suppressed and mixing between the wake and the surrounding flow is weak (e.g., Magnusson and Smedman, 1994; Hansen et al., 2012). Furthermore, wind plant wake effects are not vertically confined to the rotor area and can extend well into the planetary boundary layer, even impacting the planetary boundary layer height (PBLH, e.g., Sharma et al., 2017; Wu and Porté-Agel, 2017).

The planetary boundary layer is the region of the atmosphere influenced by the earth's surface, and its height is determined by atmospheric properties such as turbulence and temperature (Stull, 1988; Wallace and Hobbs, 2006), which can be modified by wind plants. PBLH is an important atmospheric variable governing many processes such as pollutant dispersion (Monks et al., 2009; Gan et al., 2011; Su et al., 2018; Lee et al., 2019; Miao et al., 2019) and cloud formation (Konor et al., 2009; Neggers et al., 2017), and it is a key parameter for atmospheric (Deardorff, 1972; Li et al., 2024) and wind turbine wake (Narasimhan et al., 2024) models. PBLH typically evolves over the diurnal cycle. Under daytime convective conditions, the sun warms the air near the surface, causing thermally induced convective mixing. This mixed layer is topped by a statically stable capping inversion that prevents turbulence from penetrating into the free atmosphere. The capping inversion determines PBLH under convective conditions. After sunset, the cool ground surface suppresses thermally driven turbulence, and the mixed layer decays into the residual layer. Under these stable conditions, PBLH becomes much more difficult to define, as the stable boundary layer transitions smoothly into the residual layer above it (Stull, 1988; Wallace and Hobbs, 2006).

From an experimental standpoint, the impact of wind plants on PBLH is difficult to determine for several reasons. As mentioned above, PBLH is hard to quantify under stable conditions, when wind plant wake effects are strongest. Furthermore, PBLH is influenced by multiple factors that are difficult to control for, such as terrain variations (Lieman and Alpert, 1993), surface roughness and heterogeneity (Li et al., 2018b), and synoptic patterns (Miao et al., 2019). These influences make it challenging to isolate the impacts of wind plants. In addition, wind plants can cover hundreds of square kilometers, so measurement campaigns must span large areas to capture changes induced by the turbines. For these reasons, most previous investigations into the impact of wind plants on PBLH were conducted using numerical simulations.

Numerical simulations across a wide range of fidelities have modeled the interactions between wind plants and the planetary boundary layer (Calaf et al., 2010; Wu and Porté-Agel, 2011; Stevens and Meneveau, 2017; Hsieh et al., 2021; Cheung et al., 2023; Jia et al., 2024). Representing wind turbines as momentum sinks and turbulent kinetic energy (TKE) sources within the Weather Research and Forecasting model (WRF), Fitch et al. (2012) modeled a large offshore wind plant in a neutral planetary boundary layer. They observed an increase in PBLH of 17 m (∼3 %) within the plant and a decrease of 12 m (∼2 %) in the wake. This work was extended to investigate wind plant–boundary layer interactions under a diurnal cycle (Fitch et al., 2013), showing a factor of 4 increase in PBLH within the wind plant under stable conditions and no change in PBLH under convective conditions. Downstream of the wind plant under stable conditions, PBLH was first observed to increase relative to the case without the wind plant, then decrease below the baseline as the night progressed. Lu and Porté-Agel (2015) used large-eddy simulations (LES) to investigate the impact of wind plants with different layouts on a convective planetary boundary layer,





finding a 16 % increase in PBLH over the plants for all layouts. To explore the effects of wind plants on the diurnal evolution of the planetary boundary layer at higher resolution, Sharma et al. (2017) conducted LES of the same 2-day period used by Fitch et al. (2013). Within the wind plant, they found an increase in PBLH of 175 % and 5 % under stable and convective conditions, respectively, relative to the no turbine case. The wind plant also lifted the nose height of the nocturnal low-level jet (LLJ) by ~250 m, reducing the power available to the turbines. Furthermore, the presence of the wind plant was shown to delay the transition from stable to convective conditions, then accelerate the growth of the convective boundary layer. Wu and Porté-Agel (2017) used LES to investigate the effects of temperature gradients and wind plant layout on the evolution of boundary layer flow through the plant. In all cases, the internal boundary layer generated by the wind plant deflected the planetary boundary layer upward. Under weak initial thermal stratification (1 K km$^{-1}$), this PBLH increase persisted far downstream of the plant, while stronger thermal stratification (5 K km$^{-1}$) caused PBLH to return to baseline within a few kilometers of the plant exit. By contrast, the plant layout did not have a strong effect on PBLH (Wu and Porté-Agel, 2017). Most recently, Quint et al. (2024) used WRF with the Fitch turbine parametrization to evaluate the impact of wind plants on local meteorology off the east coast of the United States over a full year. They reported increases in PBLH within the wind plant for all atmospheric stabilities, with the strongest impacts under stable conditions and when the turbines were operating below the rated wind speed. Under stable conditions, decreases in PBLH were also reported downstream of the wind plant. These studies have provided useful insights into the interactions between wind plants and the planetary boundary layer, but their findings have not yet been validated by field experiments. Furthermore, although many authors agree that PBLH increases within a wind plant under stable conditions, discrepancies exist in the magnitude of the increase, the downstream persistence, and the response under convective conditions.

The American WAKE ExperimeNt (AWAKEN), an extensive field campaign underway in the U.S. southern Great Plains region, is uniquely poised to address the question of wind plant impact on PBLH using observations (Moriarty et al., 2024). The AWAKEN campaign is the result of a multi-institutional effort to better understand wind plant–atmosphere interactions and to reduce the uncertainty of modeling tools. To this end, a region encompassing five wind plants in northern Oklahoma was equipped with in situ and remote instruments installed at 13 ground-based sites. The spatial distribution of these sites enables the comparison of atmospheric properties upstream, between rows, and downstream of the wind plants. Capitalizing on this dataset, Krishnamurthy et al. (2024) used ground-based lidars placed upstream and downstream of one of the AWAKEN wind plants to investigate the interplay between momentum flux and wind plant wake recovery. This study found that, during the frequent nocturnal LLJ events that are common in the southern Great Plains, the wind plant deflects the LLJ peak to a height above the turbines, consistent with the LES results presented by Sharma et al. (2017). In particular, Krishnamurthy et al. (2024) observed that this deflection occurs when the upstream LLJ height is below 250 m. When the LLJ is above this height, deflection does not occur because it is caused by the growth of the wind plant internal boundary layer, which remains below 250 m. This finding is especially relevant to the current investigation, as LLJ height is one of the possible definitions of PBLH under stable conditions (Liu and Liang, 2010).

The goal of the current study is to characterize the impact of wind plants on PBLH at the AWAKEN site, using both thermodynamic and turbulence-based methods. First, a thorough comparison between different methods for evaluating PBLH under





stable and convective conditions is conducted. Using the methods selected during this process, PBLH at multiple AWAKEN sites is then systematically evaluated for periods spanning from a few months to one year, depending on instrument availability. By comparing measurements taken at sites upstream and downstream of two wind plants under different atmospheric conditions, the impact of the plants on PBLH is quantified, and the extent of this impact is assessed. The layout of the manuscript is as follows: In Sect. 2, details about the AWAKEN layout and the instruments used are provided. A comparison of the PBLH calculation methods is conducted in Sect. 3. Section 4 presents the analysis of the differences in PBLH induced by the wind plant, and the effects of ambient atmospheric conditions on these differences. To conclude, Sect. 5 summarizes key findings and discusses their implications for the wind energy community.

## 2 Experimental setup and instrumentation

Instruments were deployed at the AWAKEN site starting in September 2022, and some will remain in the field through July 2025. The site of the experiment is located in northern Oklahoma, where the land use is primarily agricultural. The terrain in the region is relatively smooth, with small river valleys and ridges that cause elevation variations <100 m over ~3000 $\mathrm{km}^2$. The five wind plants in the AWAKEN domain are Chisholm View, Thunder Ranch, Breckinridge, Armadillo Flats, and King Plains, with a total installed capacity of 1.19 GW from 558 turbines (Fig. 1). In addition to the wind plants in the area, the location was chosen for its proximity to the Atmospheric Radiation Measurement (ARM) Southern Great Plains (SGP) atmospheric observatory, which has been collecting observations since 1992 (Sisterson et al., 2016).

### 2.1 AWAKEN site layout and conditions

The current investigation focuses on the instruments surrounding the Armadillo Flats (AF) and King Plains (KP) wind plants. AF includes 126 turbines, 23 of which are GE 1.72 MW turbines with 103 m rotor diameter, $D$, and 80 m hub height, $H_{\mathrm{hub}}$. Another 57 are GE 1.79 MW turbines with $D = 100$ m and $H_{\mathrm{hub}} = 80$ m. The remaining 46 are GE 2.3 MW turbines with $D = 116$ m and $H_{\mathrm{hub}} = 89$ m. KP consists of 88 GE 2.82 MW turbines with $D = 127$ m and $H_{\mathrm{hub}} = 89$ m. During most of the year, the wind blows primarily from the south, especially when the wind speed is high. In the winter, northerly winds are observed with almost the same frequency as southerlies (Krishnamurthy et al., 2021). Because of these dominant wind directions, the AWAKEN site layout was designed to measure the incoming, inter-row, and wake flows of the wind plants, especially for KP, along a north-south cross section. The primary instruments and sites used for the current investigation are shown in Fig. 1 and discussed in detail below.

### 2.2 Radiosonde

The radiosonde is considered the gold standard for atmospheric profiling because of its ability to take in situ measurements with high vertical resolution. However, its temporal resolution is limited, as most radiosondes are only launched a few times each day. In the current study, radiosonde data are primarily used to validate measurements from the infrared spectrometers, which have much higher temporal resolution but do not measure the atmospheric quantities of interest directly (see Sect. 2.3).

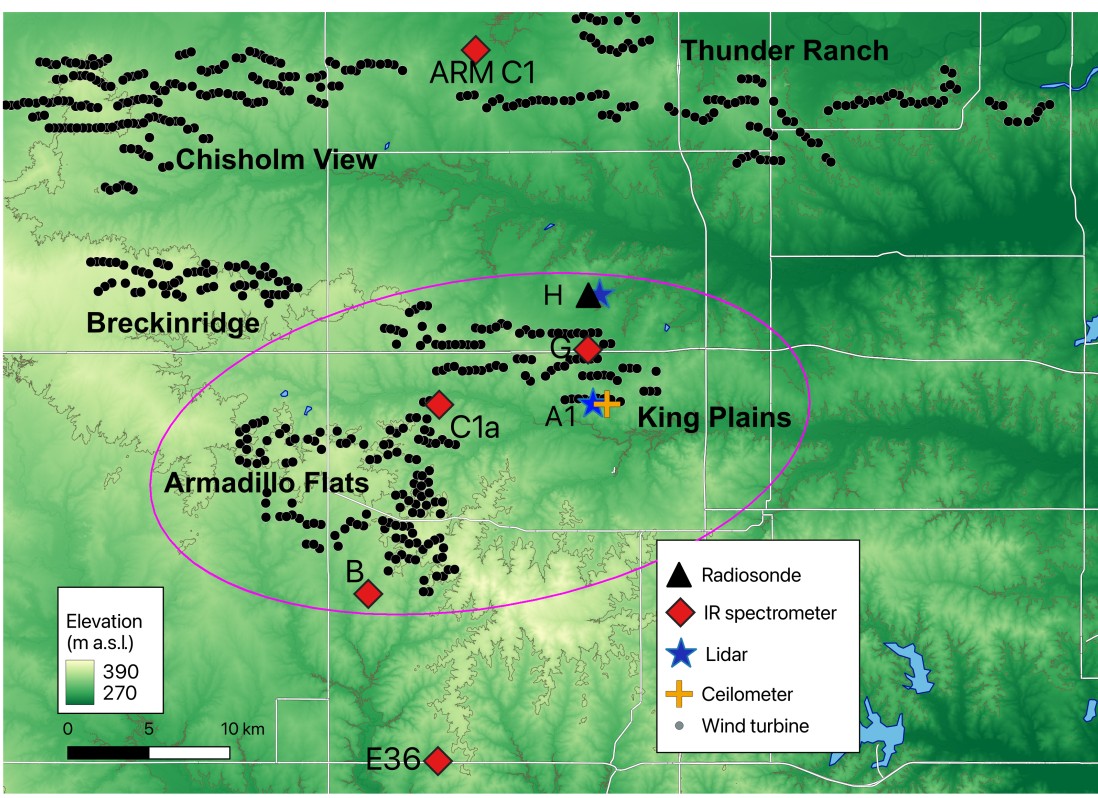

**Figure 1.** Map of the AWAKEN domain showing the five wind plants and the instrument sites used in the current investigation, including the two primary wind plants, King Plains (KP) and Armadillo Flats (AF). Terrain elevation is shown in the background (USGS, 2023). The magenta ellipse indicates the region near KP and AF that is the focus of the first part of the investigation.

The balloon-borne Vaisala radiosondes deployed at AWAKEN were launched from site H (Fig. 1) five times per day, around 02:30, 05:30, 08:30, 11:30, and 23:30 UTC, for a total of 200 launches during parts of May, July, and August 2023 (Keeler et al., 2023). They recorded moisture, pressure, temperature, and wind data at 1 Hz during each launch, which typically lasted over an hour and extended more than 20 km into the atmosphere. In the current study, only the lowest 3 km are used. It took the radiosonde ~8.5 min from launch to reach this height. During these first several minutes, the radiosondes traveled an average horizontal distance of 3 km, keeping them well away from turbines under southerly winds, and pushing them just above the northernmost row of KP when the wind was directly from the north.

## 2.3 Infrared spectrometers

Two types of infrared (IR) spectrometers are used for AWAKEN: Atmospheric Sounder Spectrometer by Infrared Spectral Technology (ASSIST) and Atmospheric Emitted Radiance Interferometer (AERI) systems. The ASSISTs used in the current





investigation are located at sites B (Letizia, 2023a), C1a (Letizia, 2023b), and G (Letizia, 2023c). The AERIs used are at the ARM SGP Central Facility, C1 (Turner, 2024), and on the Collaborative Lower Atmospheric Mobile Profiling System (CLAMPS) located at site E36 (Gebauer et al., 2023), as seen in Fig. 1. These passive, ground-based instruments measure spectrally resolved downwelling IR radiation emitted by the atmosphere. A retrieval algorithm is then used to optimally es-

timate temperature and humidity profiles from the observed spectra. The current study employs TROPoe, formerly AERIoe, which uses the optimal estimation framework based on a forward radiative transfer model that estimates the observation vector (i.e., the spectra over selected channels) from the state vector (i.e., temperature and humidity at 55 levels) (Turner and Löhnert, 2014; Turner and Blumberg, 2019; Turner and Löhnert, 2021). An optimal estimate of the state vector matching the observations is obtained through an iterative algorithm, given the nonlinear nature of the forward model. Furthermore, since the

solution of this inverse problem is intrinsically ill-defined, a priori information about the climatology of the site is required to ensure the retrieval converges to a plausible solution. For AWAKEN, ∼2000 radiosonde launches from ARM SGP were used to generate the prior dataset for each month. The retrievals from the ASSISTs (sites B, C1a, and G), the AERI at E36, and the ARM C1 AERI are generated using 30 min, 10 min, and 5 min averages, respectively.

The vertical resolution of the TROPoe retrievals starts at 6 m near the ground surface, then decreases exponentially with

elevation to 2 km at 1 km above the ground, introducing increased smoothing at higher elevations (Turner and Löhnert, 2014). This smoothing effect is captured by an A-kernel matrix that is output by the TROPoe process. The same A-kernel is applied to radiosonde profiles in order to compare them with the TROPoe retrievals and determine the impact of the vertical resolution on measurement accuracy (Blumberg et al., 2017). A qualitative example of such a comparison is presented in Fig. 2, which shows profiles measured by the radiosonde at site H and the nearest IR spectrometer at site G on 18 May 2023, along with radiosonde

profiles smoothed using the A-kernel. The profiles show good agreement, especially near the ground, which is the region most relevant for low, stable boundary layers where wind plant impacts are expected to be the strongest. Inversion layers tend to get smoothed out by the low vertical resolution of the TROPoe retrievals at higher elevations. The TROPoe algorithm also outputs a quality control (QC) flag, where a zero value indicates that the retrieval quality is acceptable. For the following analysis, retrievals with nonzero QC flag values are excluded. Retrievals are also eliminated when clouds below 3 km are detected by

the AWAKEN ceilometer at site A1 (discussed further in Sect. 2.5), as the IR spectrometer cannot capture information above the clouds (Turner and Löhnert, 2021).

## 2.4  Lidars

The scanning pulsed Doppler lidar collects simultaneous measurements of the along-beam (or radial) wind component and backscattered signal-to-noise ratio (SNR) along the laser direction, which can be used to quantify PBLH (Dang et al., 2019).

Two Streamline XR+ Doppler lidars, manufactured by Halo Photonics and deployed at sites A1 and H (Fig. 1), are used in this work for the period from 1 January 2023 to 1 January 2024 (Letizia et al., 2023a, b). The lidars are programmed to execute six-beam scans, which are post-processed to reconstruct vertical profiles of flow statistics under horizontally and temporally statistically homogeneous conditions (Eberhard et al., 1989; Sathe et al., 2015; Newsom et al., 2017; Bonin et al., 2018). The processed data files include the vertical profiles of first-order (i.e., mean wind speed and direction) and second-order (i.e.,

**Figure 2.** Comparison between temperature (top), mixing ratio (middle), and potential temperature ($\theta$, bottom) profiles measured by the radiosonde at site H and the nearby IR spectrometer at site G (see Fig. 1), including all five radiosonde launches on 18 May 2023 and the nearest IR spectrometer retrievals in time. Radiosonde profiles smoothed using the TROPoe A-kernel are included as well.

Reynolds stresses and TKE) statistics at 10 min resolution. The values of azimuth and elevation angles, together with sampling time and vertical spatial resolution, are reported in Table 1. Notably, the scanning lidar at site A1 adopts a tilted six-beam scan, since this instrument is situated within the global induction zone generated by the KP wind plant for southerly wind



**Table 1.** Scan parameters for the lidars located at sites A1 and H.

| Site | Azimuth (°) | Elevation (°) | Gate length (m) | Sampling time (s/scan) |
|------|-------------|---------------|-----------------|------------------------|
| A1 | 90, 121.8, 180, 180, 238.2, 270 | 58.3, 45, 69.5, 45, 45, 58.3 | 30 | 18 |
| H | 83.7, 155.7, 227.7, 299.7, 11.7, N/A | 45, 45, 45, 45, 45, 90 | 30 | 23 |

conditions, which violates the hypothesis of spatial homogeneity. By placing a virtual lidar at site A1 and simulating southerly flow around KP via WRF-LES simulations, Letizia et al. (2024) showed that the wind statistics retrieved by the tilted scan are

not influenced by the single-turbine induction zones related to the turbines forming the southernmost row of KP (as opposed to the regular six-beam scan), thereby partially retrieving the spatial homogeneity hypothesis required to reconstruct unbiased wind statistics. By contrast, the effects of the global wind plant blockage, extending for more than $15D$ to the south, cannot be avoided by the present scan technique (Letizia et al., 2024).

The QC of the radial wind data collected by each lidar is done based on the dynamic filter algorithm detailed in Beck

and Kühn (2017). First, instantaneous velocity samples characterized by SNR outside of the interval $[-25, 0]$ dB are discarded. Subsequently, the data are divided into bins of $[100, 100, 50]$ m in the three spatial directions and 600 s in time. For each bin, the centered bivariate probability density function of instantaneous radial wind speed and SNR is evaluated and normalized. All the occurrences with probability below a scan-dependent threshold are considered erroneous velocity reads and removed from the time records. By assuming statistical homogeneity over the scan volume at each height, mean wind components are estimated

through a least-square fit method (Päschke et al., 2015), while Reynolds stresses are obtained from the 10 min variance of the radial velocity based on geometrical arguments (Sathe et al., 2015). It is noteworthy that at this stage, the wind statistics are not corrected for the time and probe averaging of the lidar. Residual noise in the QC data has also not been accounted for, although it is expected to have a standard deviation of less than $0.1$ m s$^{-1}$ (Newsom et al., 2017). A pre-campaign comparison of the lidar-derived TKE using analogous methods showed correlation coefficients between 0.78 and 0.97 with a sonic anemometer

placed at 119 m above ground level Letizia et al. (2024), with mostly random errors and no evident biases.

## 2.5 Ceilometer

Ceilometers, operating like vertically pointing lidars, are used to obtain the cloud base height and backscatter profile of the planetary boundary layer. Continuous estimates of PBLH can then be derived from the backscatter profile (Münkel et al., 2007; Zhang et al., 2022). At AWAKEN, a Vaisala CL51 ceilometer is deployed at site A1 (Hamilton and Zalkind, 2023), alongside

one of the scanning lidars discussed in Sect. 2.4 (Fig. 1). The ceilometer measures the backscatter profile up to 15 km at 10 m spatial resolution, and the Vaisala BL-View software estimates PBLH with 16 s resolution using a merged gradient and profile fit method. The PBLH estimation algorithm outputs up to three values of PBLH at each time step. For the current investigation, the lowest value is always used.





## 3 PBLH evaluation methods

### 3.1 Thermodynamic methods

A variety of methods have been developed to identify PBLH using thermodynamic properties such as temperature and humidity, which are measured by radiosondes and IR spectrometers. In the current investigation, three of these methods are used and compared: the potential temperature gradient method, the modified parcel method, and the Heffter method. Note that the Liu–Liang method (Liu and Liang, 2010) was also explored but was ultimately not used due to the large number of theshold values that are defined for radiosonde profiles and their gradients. Because IR spectrometers introduce vertical smoothing, the Liu–Liang parameters used for radiosonde measurements would not necessarily be the same as those required for thermodynamic profiler measurements, complicating any comparison between the PBLH values obtained from the two instruments. For all methods, a maximum PBLH of 3 km is considered. A previous climatological survey of the ARM SGP site shows that most daily PBLH maxima occur within this range (Krishnamurthy et al., 2021).

### 3.1.1 Modified parcel

The parcel method defines PBLH as the height at which a parcel of air from the surface is in equilibrium with its environment, or where the virtual potential temperature ($\theta_v$) is equal to the surface value ($\theta_{v,\ \mathrm{surf}}$). The modified parcel method is used in the current investigation, which defines PBLH as the height where $\theta_v = \theta_{v,\ \mathrm{surf}} + 0.5$ K (Duncan Jr. et al., 2022). This modification is intended to improve the detection of PBLH during the evening transition (Coniglio et al., 2013). Figure 3 shows example implementations of this method for stable and convective boundary layers captured at 11:41 and 23:41 UTC, respectively, using radiosonde launches and nearby IR spectrometer retrievals. The modified parcel method accurately identifies the location of the capping inversion in the convective case. In the stable case, $\theta_v$ is constantly increasing near the surface, so the location where $\theta_v = \theta_{v,\ \mathrm{surf}} + 0.5$ K occurs just above the surface.

### 3.1.2 Potential temperature gradient

The potential temperature gradient method, as described by Duncan Jr. et al. (2022), defines PBLH as the location of the maximum vertical gradient of potential temperature, $d\theta/dz$, where $z$ is the height above ground level. The bottom portion of the IR spectrometer profile is excluded from the PBLH search due to nonphysical temperature gradients observed in the near-surface portion ($z < 100$ m) of some retrievals, as implemented by Blumberg et al. (2017). In the stable example shown in Fig. 3, the potential temperature gradient method detects a PBLH value just above the surface from the radiosonde and at the first level above $z = 100$ m from the IR spectrometer. In the convective case, the method accurately detects the bottom of the capping inversion from the radiosonde profile, but appears to overestimate PBLH from the IR spectrometer retrieval relative to the radiosonde. This overestimate is due to the vertical smoothing introduced by the retrieval, which shifts the height of the maximum gradient upwards.



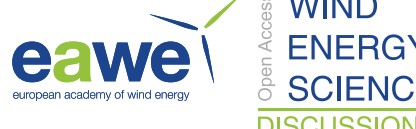

**Figure 3.** Profiles obtained from radiosonde soundings at site H and IR spectrometer retrievals at nearby site G, demonstrating the implementation of the modified parcel, potential temperature gradient, and Heffter methods for determining PBLH. The top panels show a stable example and the bottom panels show a convective example. Line color indicates instrument while line style denotes quantity of interest.

### 3.1.3 Heffter

The Heffter method (Heffter, 1980) uses potential temperature gradients to identify inversion layers and potential temperature differences across these layers to determine their strength (Sivaraman et al., 2013). When applying this method to a radiosonde sounding, the profile is first smoothed using a moving average window of 32 points, corresponding to ~180 m. For both radiosonde and IR spectrometer profiles, inversion layers are identified as segments where the vertical gradient, or lapse rate,





of the smoothed potential temperature ($d\tilde{\theta}/dz$, where the tilde denotes smoothing) profile is greater than $0.001$ K m$^{-1}$. Note
that other implementations of the Heffter method have used a critical lapse rate of $0.005$ K m$^{-1}$, but this threshold was
originally intended for marine boundary layers (Heffter, 1980). Delle Monache et al. (2004) found that a critical lapse rate
of $0.001$ K m$^{-1}$ is more appropriate for the land-locked ARM SGP site. Next, the potential temperature difference across
each detected inversion layer is calculated. The lowest inversion layer with a potential temperature change of at least 2 K is
selected. Within this layer, PBLH is defined as the lowest point where the potential temperature is 2 K greater than the bottom
of the layer. If no inversion layer meeting these criteria is identified, PBLH is defined as the height of the maximum potential
temperature gradient (as in the potential temperature gradient method described above). Figure 3 shows the implementation of
the Heffter method, including the detected inversion layers. In the examples provided, this method identifies the most consistent
values of PBLH between the IR spectrometer and radiosonde profiles. In the convective case in particular, the location of the
capping inversion is identified in the profiles from both instruments, despite the smoothing introduced by the IR spectrometer.

### 3.1.4 Thermodynamic method comparison and selection

Because the radiosonde is considered the gold standard for atmospheric profiling measurements, good agreement between
PBLH detected by the IR spectrometer and the radiosonde using a particular method would signify the reliability of the method
for further PBLH analysis with the IR spectrometers. Therefore, the statistical agreement between the results obtained using IR
spectrometer retrievals and radiosonde soundings for each method is explored. For each sounding collected by the radiosonde at
site H, the nearest retrieval (in time) captured by the IR spectrometer at site G is identified. After removing periods with clouds
and nonzero QC flags, as described in Sect. 2.3, and limiting the time between the sounding and the corresponding retrieval to
less than 15 minutes, 67 pairs of profiles remain ($N = 67$). Note that some differences between the profiles collected by the
radiosonde and the IR spectrometer are expected due to the spatial offset between the two instruments, including the fact that
the IR spectrometer is stationary while the radiosonde moves with the wind.

PBLH is computed from each dataset using all three thermodynamic methods. Figure 4 presents the results of these com-
parisons. Of the three methods, the Heffter method shows the best agreement between PBLH obtained from the radiosonde
and PBLH from the IR spectrometer. The coefficient of determination ($R^2$), obtained from a linear least-squares regression of
the two sets of PBLH estimates, is 0.95 when the Heffter method is used, but 0.80 and 0.71 for the modified parcel and po-
tential temperature gradient methods, respectively. The IR spectrometer detects lower PBLH values than the radiosonde using
all methods, indicated by a linear regression slope $< 1$ with most points below the $y = x$ line, due to the vertical smoothing
introduced by the IR spectrometer data retrieval. This underestimate is most pronounced when the modified parcel method
is used. With a regression slope of 0.76, the IR spectrometer-based potential temperature gradient and Heffter methods both
underestimate PBLH by similar amounts in an average sense. However, under stable conditions, the potential temperature
gradient method consistently detects PBLH values just above the minimum 0.1 km level, regardless of the radiosonde-based
estimate. The modified parcel method also appears to detect very low values of PBLH under stable conditions. The Heffter
method, on the other hand, estimates a wider range of PBLH values, making wind plant-induced changes in PBLH easier to
detect. Furthermore, the observed robustness of the Heffter method is consistent with the findings of Jozef et al. (2022), who





showed that the Heffter method produced the best agreement between automatically detected and manually identified PBLH values under stable conditions. The Heffter method is also one of the well-established methods used by ARM to produce their
PBLH value added product (VAP) from radiosonde retrievals (Sivaraman et al., 2013). Figure 5 shows a sample time series of PBLH estimates over 10 days from both instruments using the Heffter method. A couple of the PBLH estimates from the radiosonde under convective conditions show higher values than those from the IR spectrometer, consistent with the scatter plot in Fig. 4. However, the overall agreement between the two instruments is good, especially under stable conditions. Based on the above discussion, the Heffter method is selected as the thermodynamic method for further PBLH analysis throughout
the rest of the study.

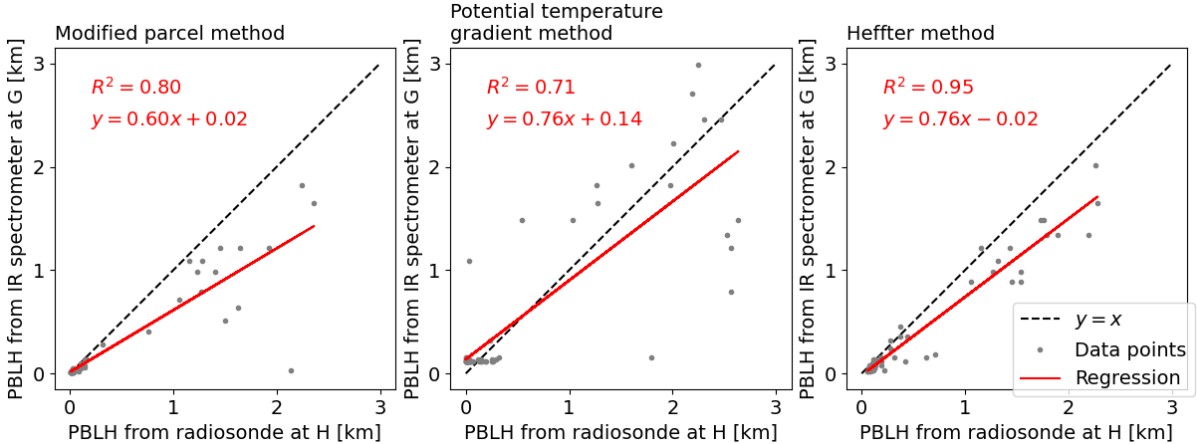

**Figure 4.** Comparison between PBLH computed using the radiosonde soundings from site H and the IR spectrometer retrievals from nearby site G, for three different methods, with $N = 67$. For each method the linear least-squares regression is shown, along with the coefficient of determination ($R^2$) and the $y = x$ line for reference.

## 3.2 Turbulence-based method

We also adopt a turbulence-based method to quantify PBLH across varying atmospheric stability conditions from lidar data. Canonically, turbulence within the planetary boundary layer is generated at the ground due to the mean wind shear and trans-ported vertically by turbulent mixing. Since the free atmosphere generally exhibits a low turbulence level, several turbulence-
related statistics have been proven to sharply decrease above the planetary boundary layer, thus defining the latter as the volume of the atmosphere where turbulence is present (Stull, 1988; Garratt, 1994). Examples of these statistics, all measurable via ground-based Doppler lidars, are the mean backscatter coefficient (Baars et al., 2008), the backscatter ratio (Kong and Yi, 2015), and the horizontal (Pichugina and Banta, 2010) and vertical velocity variances (Vickers and Mahrt, 2004; Tucker et al., 2009; Dai et al., 2014; Berg et al., 2017). The horizontal and vertical velocity fluctuations, in particular, are expected to decrease
from the ground down to a minimum value at the PBLH (Dai et al., 2014), and the vertical velocity variance is considered to be the quantity that best captures the definition of PBLH (Tucker et al., 2009). Therefore, in this work the minimum of the vertical





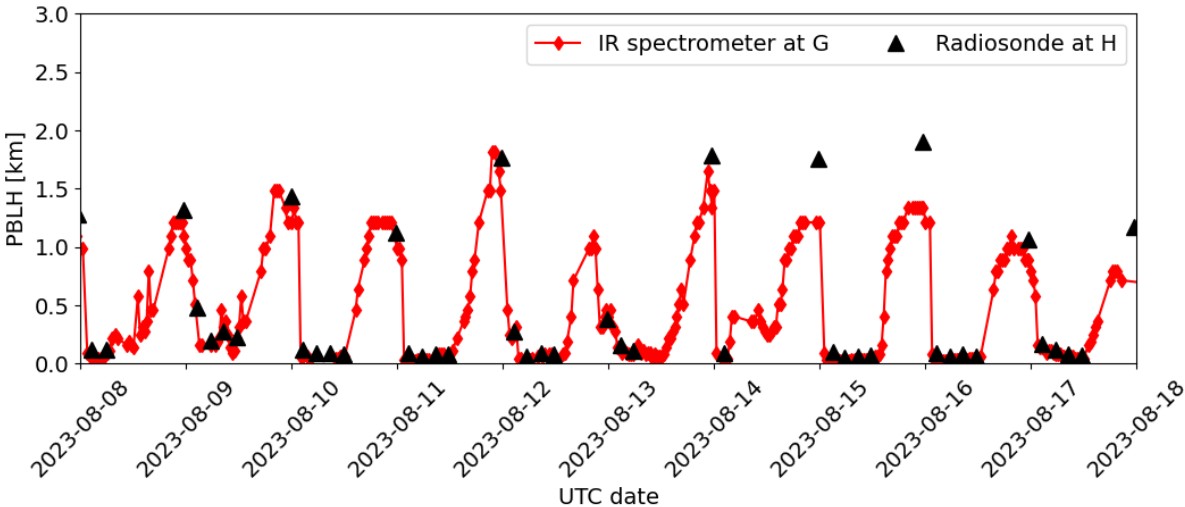

**Figure 5.** Time series of PBLH from the IR spectrometer retrievals at site G and the radiosonde soundings at site H, both computed using the Heffter method.

velocity variance profile ($\overline{w'w'}(z)$) is used to quantify PBLH from the lidars. Puccioni et al. (2024) demonstrated the efficacy of this method under stable conditions, showing consistent results with historical PBLH data recorded at the ARM SGP site. It is also a more generalized version of the method employed by Tucker et al. (2009), which uses an empirically determined threshold of $\overline{w'w'}(z)$ to estimate PBLH under convective and stable conditions when PBLH is above the blind zone of the lidar.

Using the minimum of $\overline{w'w'}(z)$ rather than a threshold introduces operative challenges in implementing automatic PBLH identification due to the limited vertical resolution of the lidar (which may not suffice to resolve all the vertical turbulent motions) and the finite sampling time (which adds statistical uncertainty to the second-order vertical profile). A procedure is here adopted to mitigate the false detection of local minima due to instrumental and statistical noise in the estimation of the height of minimum $\overline{w'w'}(z)$ (i.e., the PBLH), as visualized in Fig. 6. The initial $\overline{w'w'}(z)$ profile (gray line) is bin-averaged every $\Delta z = 100\,\text{m}$ (black symbols) between $120\,\text{m}$ and $3000\,\text{m}$, and the minimum of the bin-averaged profile ($z_*$) is identified (blue square symbol). This value is not the true PBLH, but it is expected to be close to it. Thus, after identifying $z_*$, the PBLH is found as the minimum of $\overline{w'w'}(z)$ only within the interval $z_* \pm \Delta z/2$ (blue cross symbol in Fig. 6).

### 3.3 Thermodynamic and turbulence-based method comparison

Thermodynamic and turbulence-based PBLH evaluation methods use different quantities to attempt to capture the same atmospheric processes. Here, the agreement between these different approaches is explored and compared, using the instruments at the two sites just south (typically upstream) of the wind plants (sites A1 and B), alongside the PBLH estimates from the ceilometer at site A1. Figure 7(a) shows a sample time series of PBLH calculated using each method, including several diurnal cycles. Note that the lidar and ceilometer measurements are downsampled and smoothed to match the 30 min resolution of





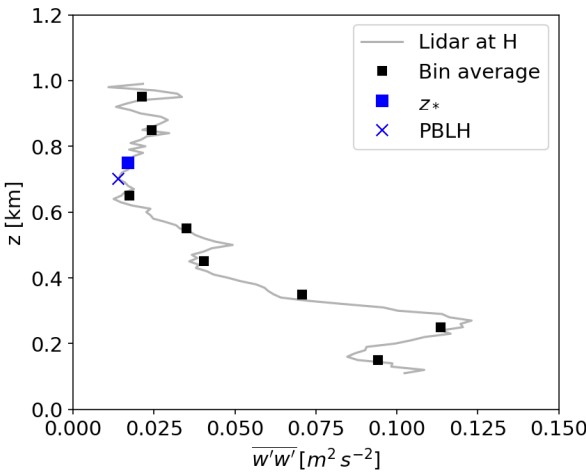

**Figure 6.** Example of PBLH quantification based on the profile of vertical velocity variance ($\overline{w'w'}$) retrieved from the scanning lidar at site H.

the IR spectrometer. All methods successfully capture the morning and evening transitions, as well as day-to-day variations in PBLH (e.g., 2023-08-27 and 2023-09-05). Under nighttime stable conditions, the turbulence-based method using data from the lidar generally overestimates PBLH relative to the thermodynamic method using data from the IR spectrometer, while the ceilometer-based estimates typically fall somewhere in between. Under daytime convective conditions, the lidar also tends to detect higher PBLH values than the IR spectrometer. The ceilometer tends to follow the evolution measured by one of the

other two instruments, though it sometimes records a smaller or larger value than either. The other panels of Fig. 7 compare the results of the three instruments directly. In Fig. 7(b) and (c), most of the data points are above the $y = x$ line, reflecting the higher estimates of PBLH from the lidar relative to the other two instruments. Panels (b) and (d) show the lower estimates of PBLH from the IR spectrometer under stable conditions, i.e., when PBLH $< 1$ km. The comparison between PBLH from the lidar and ceilometer at site A1 appears the most linear (Fig. 7c), consistent with the similarity in measurement principles

of the two instruments, as well as their co-location. The relationship between PBLH calculated using the three instruments is quantified using Spearman's rank correlation coefficient (Hauke and Kossowski, 2011), rather than the linear regression presented in Fig. 4, as the different methods use different quantities that may relate nonlinearly. The PBLH estimates from the IR spectrometer and lidar show the strongest relationship, with a correlation coefficient of $\rho = 0.70$, indicating that the results of the two methods tend to evolve together, but there are some important differences. These differences are not unexpected,

as the two instruments are located 20 km apart and measure distinct atmospheric properties. On the other hand, the general agreement observed between the two methods, as well as their agreement with the ceilometer, an instrument widely used for PBLH estimation, provides confidence in their use for the analysis of wind plant effects on PBLH. Furthermore, the following analysis compares PBLH measurements taken by the same type of instrument at different sites, so one-to-one correspondence between different instruments is not required.

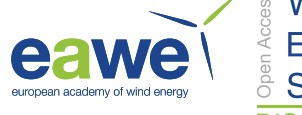



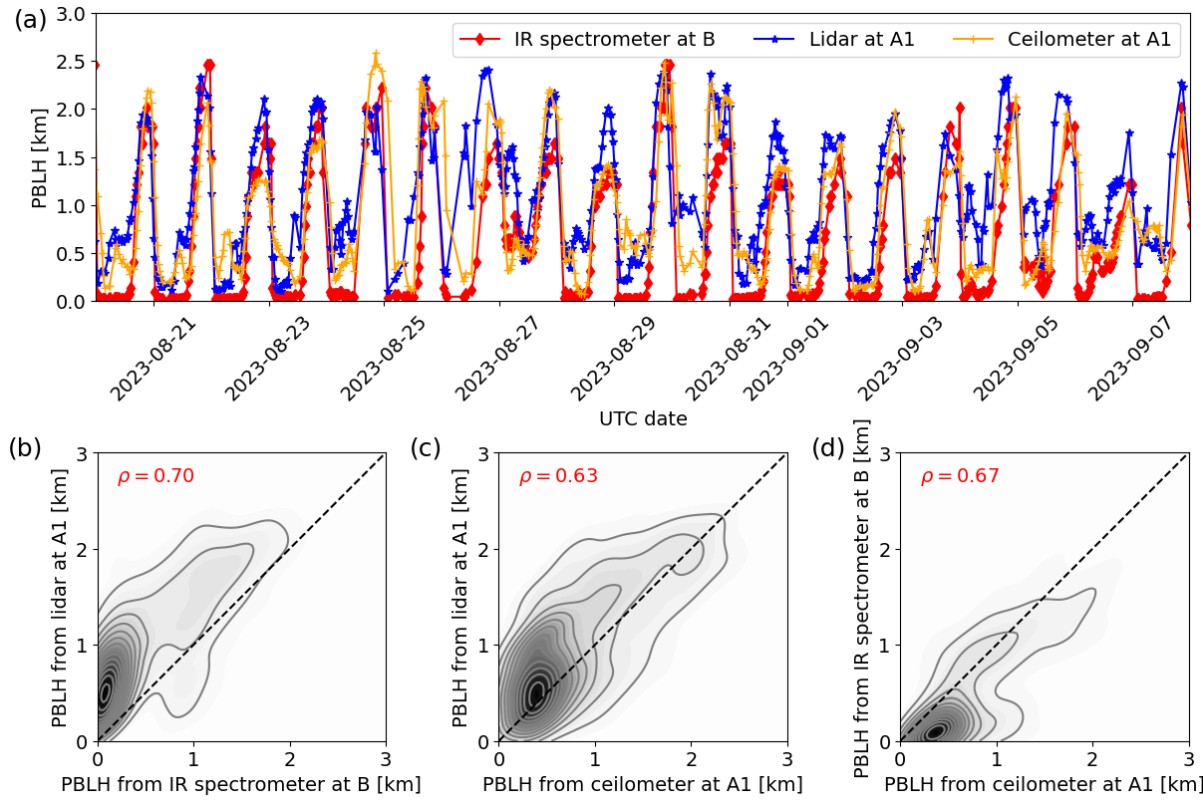

**Figure 7.** (a) Time series of PBLH computed using the IR spectrometer retrievals at site B with the Heffter method, the scanning lidar at site A1, and the ceilometer at site A1. (b,c,d) Kernel density estimates of measurements from each pair of instruments, with Spearman's rank correlation coefficient, $\rho$, shown in the panels, and $N = 2302$.

## 4 Wind plant wake effects on PBLH

The PBLH detection methods detailed in Sect. 3 are now applied to profiles acquired upstream and downstream of KP and AF to quantify the wind plants' impact on PBLH, using both thermodynamic and turbulence-based methods. We start by focusing on the locations in the immediate vicinity of the two wind plants, indicated by the ellipse in Fig. 1, before exploring the effects at more distant sites (Sect. 4.5). Note that the radiosonde and ceilometer are not used for this analysis, as each is only located on one side of the wind plants, preventing direct comparison between upstream and downstream locations.

### 4.1 Data sampling

The period used in the following analysis is between 16 May 2023 and 16 October 2023 for the IR spectrometers and between 1 January 2023 and 1 January 2024 for the lidars, based on data availability. PBLH measurements from the IR spectrometers at sites B, C1a, and G, and the lidars at sites A1 and H are binned by wind direction ($\phi$) and atmospheric stability. For the





IR spectrometers, the wind direction used for binning is obtained from the scanning lidars operating in profiling mode at sites A1, A2 (another location south of KP), or H, depending on which site is outside of the turbine wakes, using the point nearest to the wind turbine hub height. If multiple sites are unwaked, their measurements are averaged. In the following analysis, periods are separated into southerly ($160° < \phi \le 200°$) and northerly ($\phi > 340°$ or $\phi \le 20°$) wind conditions. For lidar-based analysis, southerly (northerly) wind conditions are defined as periods when the rotor-averaged wind direction measured by the

upstream lidar at site A1 or H are within the same ranges listed above. Periods where the difference in wind direction between sites A1 and H is greater than $20°$ are removed from the lidar-based analysis. Site locations relative to the wind plants (i.e., upstream, wind plant influenced, and far downstream) are listed in Table 2. Note that when the wind is from the north, site G is downstream of the first row of KP. However, previous studies (e.g., Wu and Porté-Agel, 2017) have shown that the wind plant boundary layer grows with downstream distance from the wind plant entrance, so the influence of the turbines is expected to

be much less at site G than at C1a or B, and site G is considered upstream.

**Table 2.** Locations of sites in Fig. 1 relative to the wind plants under southerly and northerly wind directions.

| Wind direction | Upstream sites | Wind plant influenced sites | Far downstream site |
|---|---|---|---|
| Southerly ($160° < \phi \le 200°$) | A1, B | C1a, G, H | ARM C1 |
| Northerly ($\phi > 340°$ or $\phi \le 20°$) | G, H | A1, B, C1a | E36 |

The Obukhov length, $L$, is used to determine atmospheric stability. Stability classes are delineated by ranges of $L$, defined in Table 3 and based on the definitions provided by Krishnamurthy et al. (2021). Measurements from four surface flux stations within the AWAKEN domain are used to compute $L$ with 30 min resolution (Pekour, 2024a, b, 2023a, b). Three of the surface flux stations are adjacent to shipping containers supporting other instrumentation, so their measurements are compromised

under certain wind directions, as detailed by the readme file for each sensor (Pekour, 2024a, b, 2023a, b). Furthermore, each station has different periods of data availability throughout the campaign. Therefore, the wind direction and date are used along with the instrument's QC flag to determine which surface flux station is used to obtain $L$ at any given time. An additional common-sense sanity check is applied: Periods when values of $L$ are out of sync with time of the day (i.e., strongly stable conditions, $1/L > 0.02$ m$^{-1}$, between sunrise and sunset, or strongly convective values, $1/L < -0.01$ m$^{-1}$, between sunset

and sunrise), are flagged as invalid. If multiple measurements are valid, one is selected, with highest preference given to the station south of the wind plants with the longest period of data availability, and lowest preference given to the station north of the wind plants. If none of the surface flux stations measure valid values of $L$ during a given 30 min period, that period is excluded from further analysis.

Because the lidar data spans all of 2023 and only one surface flux station was deployed for that entire duration, UTC hour

is used as a proxy for stability for the lidar-based PBLH analysis. The distributions of stability classes for each UTC hour, obtained during the period used for the thermodynamic profiler-based analysis, is presented in Fig. 8. These distributions show that the boundary layer is stable at least 91 % of the time between 01–11 UTC (20–06 local time, UTC−5), and is convective




**Table 3.** Definition of stability classes based on Obuhkov length, $L$, derived from AWAKEN surface flux station measurements, based on the definitions provided by Krishnamurthy et al. (2021).

| Stability | Obukhov length ($L$) |
|---|---|
| Stable | $0\text{ m} < L \leq 500\text{ m}$ |
| Near-neutral | $|L| > 500\text{ m}$ |
| Convective | $-500\text{ m} < L \leq 0\text{ m}$ |

at least 84 % of the time between 14–21 UTC. Therefore, for the lidar-based analysis, PBLH values measured between 01-11 UTC and values measured between 14–21 UTC are binned together as stable and convective, respectively.

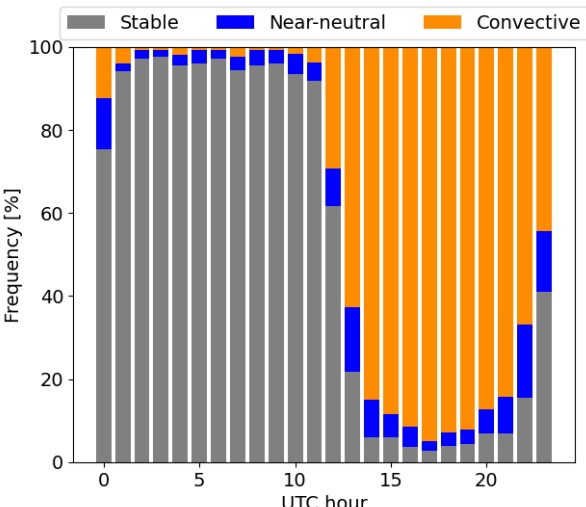

**Figure 8.** Distribution of stability classes for each hour of the day, defined by the Obukhov length obtained from surface flux stations, as described in Table 3.

## 4.2 Effects of atmospheric stability on wind plant impacts

Wind plant impacts on PBLH are first investigated for stable conditions, where previous studies have observed the strongest effects (Fitch et al., 2013; Sharma et al., 2017; Quint et al., 2024). Figure 9 shows PBLH distributions from each of the three IR spectrometers for stable periods, as defined in Table 3, separated by wind direction. When the wind is from the south (Fig. 9(a)), PBLH values at site B (upstream) tend to be smaller than those at sites C1a and G (downstream). The significance of this difference is tested using the two-sample Kolmogorov–Smirnov test, which provides the likelihood that two samples come from the same distribution (called the null hypothesis), $p$. The null hypothesis is rejected if $p < 0.05$. Under stable conditions and southerly winds, with a total of $N = 258$ data points from all three IR spectrometers, $p < 0.05$ when comparing the distributions from sites B and C1a, as well as the distributions from sites B and G. When comparing sites C1a and G,





$p = 0.94$. These $p$-values show that PBLH measured at site B is significantly different from that measured at sites C1a and

G, while PBLH measurements from C1a and G are likely to be from the same distribution. The similarity between the PBLH distributions at sites C1a and G is particularly noteworthy considering the differences between the wind plants upstream of the two sites. The KP wind plant upstream of G is arranged in rows, while the AF wind plant upstream of C1a has a more random layout. Furthermore, site G has just a few rows of turbines upstream, while site C1a has dozens of turbines to the south. The similarity in PBLH distributions at the two sites suggest the impact of the wind plant on PBLH plateaus after a certain

downstream distance after entering the plant.

Although these results suggest that the wind plant is responsible for the observed increase in PBLH, the differences between the sites could be caused by differences in terrain or surface roughness that may affect PBLH. Therefore, measurements taken during northerly winds are used to isolate the impacts of the wind plant. As shown in Fig. 9(b), PBLH measured at site G (upstream) tends to be lower than that measured at sites B and C1a (downstream) when the wind is from the north. With only

81 data points in the northerly wind stable condition sample, none of the computed $p$ values are low enough to reject the null hypothesis. However, combined with the trends observed for southerly winds, these results provide further evidence that wind plants increase the height of the planetary boundary layer under stable atmospheric conditions.

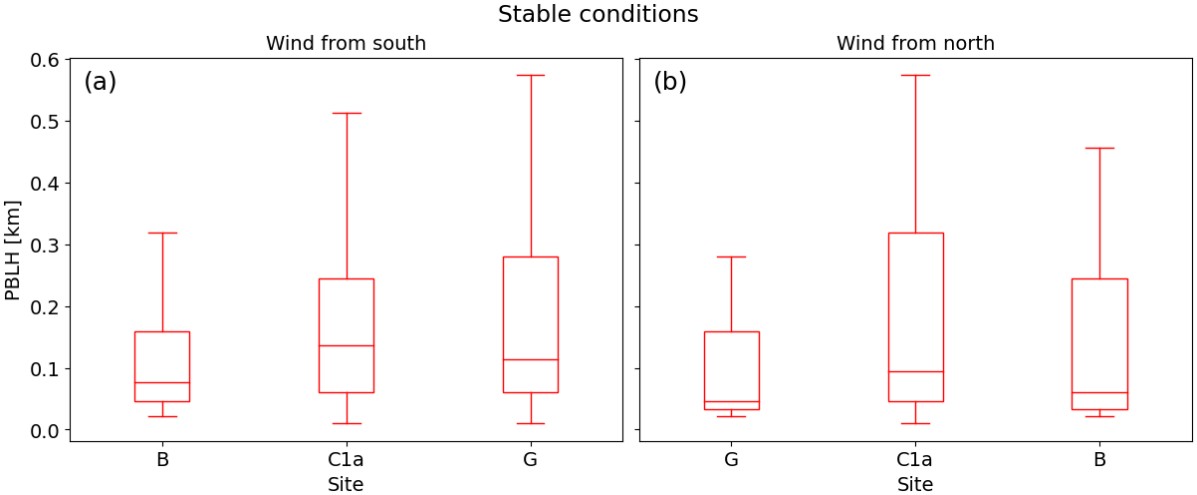

**Figure 9.** Box and whisker plots of PBLH measured by the three IR spectrometers nearest the AF and KP wind plants under stable conditions, when wind is (a) from the south ($160° < \phi \leq 200°$), with $N = 258$, and (b) from the north ($\phi > 340°$ or $\phi \leq 20°$), with $N = 81$. The boxes are bound by the first and third quartile of the distribution. The lines inside the boxes indicate the median. The whiskers extend to 1.5 times the interquartile range. Outliers are not shown for clarity. Sites on the horizontal axis are arranged from upstream to downstream for each wind direction.

PBLH distributions measured by the lidars at sites A1 and H under stable conditions (01–11 UTC) are shown in Fig. 10 for (a) wind from the south and (b) wind from the north. These distributions show consistent trends with those measured

by the lidars are larger, the upstream site exhibits lower




values of PBLH for both southerly and northerly winds. Under southerly winds, the difference in PBLH distributions between sites A1 and H is clear. Under northerly winds, the difference is weaker, with the upper bound of the box (third quartile) and the upper whisker of the distribution at site H extending above those at site A1. These directional differences suggest some spatial variability in PBLH that is not caused by the wind plant. Previous studies have emphasized the importance

of surface heterogeneity on the structure of the planetary boundary layer, particularly at the ARM SGP site (Acevedo and Fitzjarrald, 2001; Wharton et al., 2014; Krishnamurthy et al., 2021; Huang et al., 2022). As the regional land use is dominated by agriculture, each land owner plants different crops, which also vary throughout the year, influencing surface roughness, soil moisture, and energy fluxes. In addition, for northerly flow the turbulence profiling by the lidar at site A1 is expected to be affected by the presence of strong individual turbine wakes, which are known to cause significant inhomogeneities in the

flow (Lundquist et al., 2015). The accuracy of the lidar profiling in this condition has not being characterized, and the smaller increase in PBLH at site A1 may be a consequence of the resulting uncertainties in $\overline{w'w'}$. Still, the lower bound of the box (first quartile) at site H is well below that at site A1, and the median PBLH value at site H (0.46 km) is less than that at site A1 (0.54 km), indicating that the wind plant is causing an upward deflection of the PBLH that is strong enough to be observed over these other effects. For both southerly and northerly winds, $p < 0.05$ when comparing the lidar-obtained PBLH distributions at

sites A1 and H.

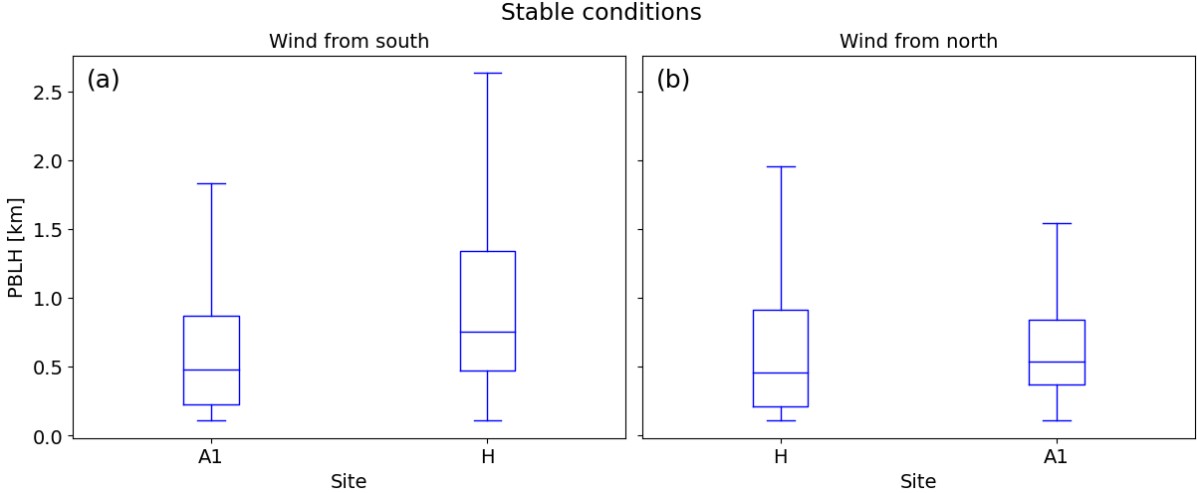

**Figure 10.** Box and whisker plots of PBLH measured by the two lidars under stable conditions, when wind is (a) from the south, with $N = 2099$, and (b) from the north, with $N = 809$. Sites on the horizontal axis are arranged from upstream to downstream for each wind direction.

Figure 11 presents the same analysis for convective conditions, as defined in Table 3 using data from the IR spectrometers. Visually, there is no clear difference between the PBLH distributions measured at the three sites when the wind is from the south or north. Correspondingly, all $p$-values computed are larger than 0.05, confirming the lack of significant differences between the distributions.





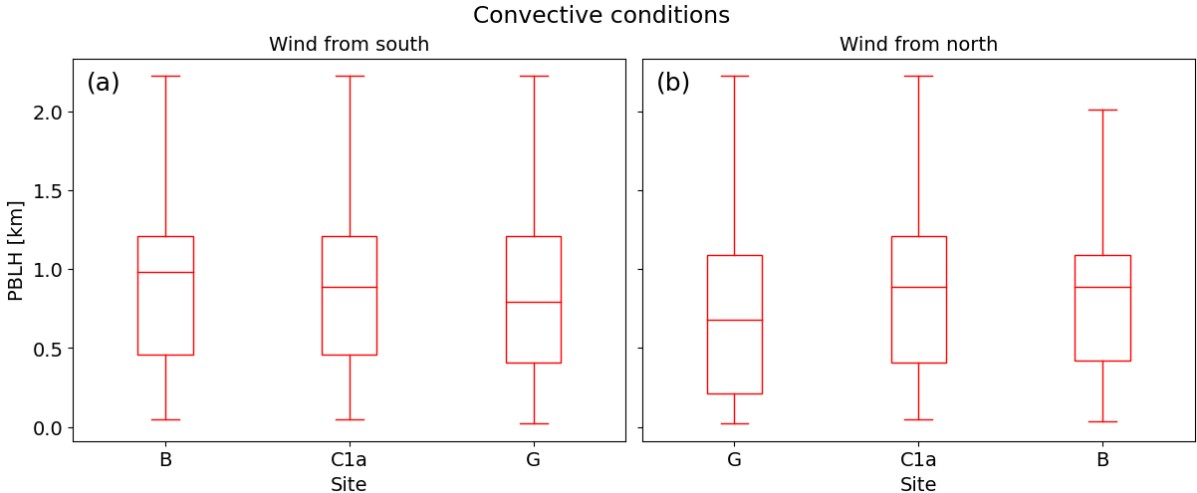

**Figure 11.** Box and whisker plots of PBLH measured by the three IR spectrometers under convective conditions, when wind is (a) from the south, with $N = 121$, and (b) from the north, with $N = 90$.

For convective conditions, the distributions of PBLH measured by the lidars also show consistent findings with those measured by the IR spectrometers (Fig. 12). Under both southerly and northerly winds, the PBLH distributions at sites A1 and H are very similar, with $p > 0.05$. These findings are consistent with the results of simulations, which showed little or no change in PBLH under convective conditions (Fitch et al., 2013; Sharma et al., 2017).

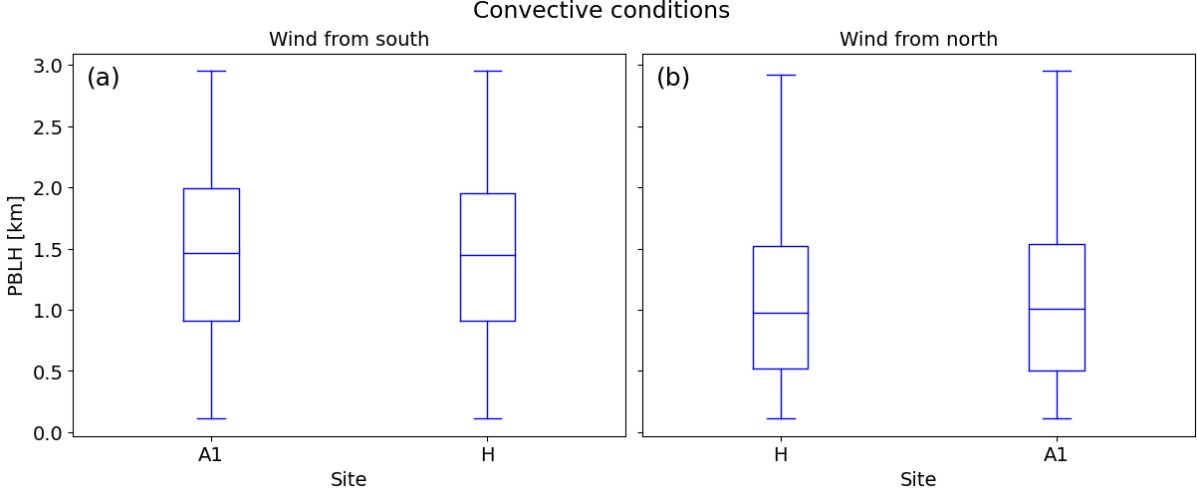

**Figure 12.** Box and whisker plots of PBLH measured by the two lidars under convective conditions, when wind is (a) from the south, with $N = 1231$, and (b) from the north, with $N = 638$.



## 4.3 Diurnal variability of wind plant impacts

In Sect. 4.2 we demonstrate the strong influence of atmospheric stability, tied to diurnal variability, on wind plant impacts on PBLH. Here, we explore further hourly variations that are missed when categorizing by stability alone. Figure 13(a) compares PBLH distributions from the three IR spectrometers under southerly wind conditions, now binned by hour of the day. Northerly conditions are not included here due to lack of sufficient data points. Diurnal variations are much larger than differences between sites. During the day (∼12 UTC to ∼01 UTC), PBLH is distributed around 1.5 km at all three sites. No clear differences

are observed between these sites, consistent with the results presented in Fig. 11. At night (∼01 UTC to ∼12 UTC), PBLH is much lower, typically less than 0.5 km. Some differences can be observed between the sites, but they are difficult to detect given the low values of PBLH. To highlight these variations, Fig. 13(b) shows the difference between PBLH measured at the downstream sites (C1a and G) and the upstream site (B), normalized by the upstream PBLH ($\frac{\mathrm{PBLH_{downstream}} - \mathrm{PBLH_{upstream}}}{\mathrm{PBLH_{upstream}}} \times 100\%$). As the planetary boundary layer flow passes through the wind plant, the presence of the turbines consistently increases PBLH

by 20–50 % during the nighttime hours.

Maximum PBLH differences occur around 01 UTC (08 local time) and 07–08 UTC (02–03 local time). The strong increase around 01 UTC suggests that the presence of the wind plant delays the evening transition from convective to stable conditions. This delay is attributed to added mixing, contributed by the wind turbines, which counteracts the suppression of turbulence by the cooling surface. The second maximum relative PBLH change occurs in the middle of the night, when the upstream

boundary layer, measured at site B, is very shallow. The effects of upstream PBLH will be explored further in Sect. 4.4.

Differences between PBLH measured by the lidar at the upstream (A1) and downstream (H) sites are stronger relative to diurnal variations than for the IR spectrometer measurements. In Fig. 14(a), the PBLH distribution at site H is consistently higher than that at site A1 throughout the night, from 0–13 UTC. Figure 14(b) presents the percent change in PBLH between sites A1 and H, showing median increases of 20–110 % throughout the night, which are larger but on the same order of

magnitude as the increases measured by the IR spectrometers. The lidars likely detect a stronger plant-induced PBLH increase compared to the IR spectrometers due to their different measuring principles. Turbine wakes inject additional turbulence into the ABL, which then causes more vigorous mixing of mean flow quantities, including temperature. Previous studies have found wind plant-induced increases in TKE up to 30 % (Bodini et al., 2021), but temperature changes of just 1–2 K at most (Rajewski et al., 2013). As the lidars actively measure the vertical velocity variance, they are likely more sensitive to wake effects on

PBLH. Conversely, the IR spectrometers may have more difficulty sensing a few degrees of warming or cooling caused by the wind plant, especially in light of the significant vertical smoothing occurring at high altitudes. The lidar-measured PBLH increases at the uppermost whisker of the distribution can be as high as 800 %, significantly larger than the maximum of 300 % captured by the IR spectrometers. This increased variability is related to the larger range of PBLH values measured by the lidars under stable conditions.

The lidars detect the largest wind plant-induced change in PBLH in the early hours of the night, around 01 UTC, providing additional evidence for a wind plant-induced delay in evening boundary layer transition. The magnitude of the change then decreases throughout the night until it is no longer detectable after sunrise at 14 UTC. This trend also obeys an inverse rela-




**Figure 13.** (a) Box and whisker plots of PBLH measured by the three IR spectrometers, binned by UTC hour, when wind is from the south. (b) Distributions of the difference between PBLH measured at the downstream sites (C1a and G) and the upstream site (B) when wind is from the south, normalized by the upstream PBLH and binned by UTC hour. The gray and orange boxes indicate the periods where stable and convective atmospheric conditions, respectively, are recorded at least 80% of the time, as presented in Fig. 8.

tionship with the lidar-measured upstream PBLH magnitude, where turbulence-based PBLH is shallowest at the beginning of the night, then deepens as the night progresses.

**4.4 Effects of upstream PBLH on wind plant impacts**

The results discussed in the previous sections suggest that the magnitude of the impact of wind plants on PBLH in the wake depends on the value of PBLH upstream of the plant. We now explore this relationship directly by binning the relative PBLH





**Figure 14.** (a) Box and whisker plots of PBLH measured by the two lidars, binned by UTC hour, when wind is from the south. (b) Distributions of the percent difference between PBLH measured at the downstream site (H) and the upstream site (A1) when wind is from the south, normalized by the upstream PBLH and binned by UTC hour. The gray and orange boxes indicate the periods where stable and convective atmospheric conditions, respectively, are recorded at least 80 % of the time, as presented in Fig. 8.

difference between the downstream sites and the upstream site by upstream PBLH. In Fig. 15(a), PBLH measured by the IR spectrometer at site B is used to bin the relative difference in PBLH measured at sites C1a and G, using a bin width of 0.15 km, when wind is from the south. The relative PBLH increase at both sites C1a and G is largest for lower values of upstream PBLH, and drops off quickly as upstream PBLH increases. Once PBLH at site B is above 0.3 km, the impact of the wind plant on downstream PBLH is negligible. This dependence on upstream PBLH is in line with the trends observed by Krishnamurthy et al. (2024) regarding wind plant impacts on LLJ nose height, which is another metric sometimes used to define PBLH (Liu and Liang, 2010). Krishnamurthy et al. (2024) showed that the LLJ is elevated downstream of a wind plant, but only when



the upstream LLJ nose height is below 0.25 km. In the current investigation, we apply the same 0.25 km separation to the upstream PBLH, as defined using the IR spectrometers. Figure 15(b) presents the relative PBLH difference, split into cases where PBLH at site B is less than and greater than 0.25 km. A clear separation emerges, where cases with lower upstream PBLH see a median increase in PBLH of 39 % at site C1a and 33 % at site G, while cases with higher upstream PBLH see a 0 % median increase at both sites. When PBLH at site B is less than 0.25 km, the entire interquartile range of the PBLH

difference distributions at both C1a and G are above 0.

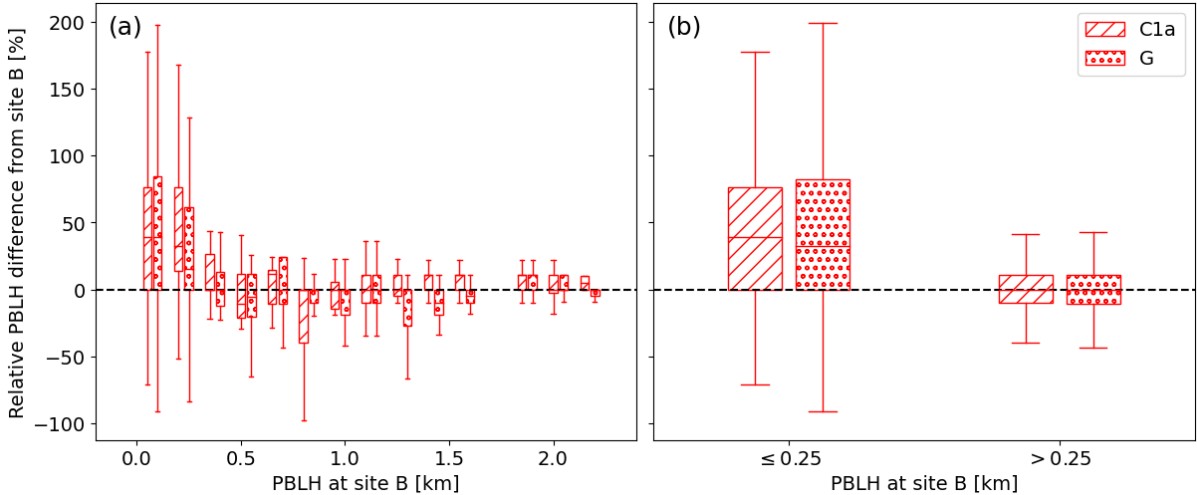

**Figure 15.** Box and whisker plots of the percent difference between PBLH measured by the IR spectrometers at the downstream sites (C1a and G) and the upstream site (B) when wind is from the south, normalized by the upstream PBLH. The distributions are binned by upstream PBLH (a) with bin widths of 0.15 km and (b) separated into values above and below 0.25 km.

Figure 16 presents the same analysis using the lidar data. Once again, the turbulence-based method for determining PBLH reveals the same trends as those observed using the thermodynamic method, just with greater magnitude. For the lowest values of upstream PBLH, the median increase in downstream PBLH is larger than 100 %. Because of the blind zone of the lidar, which extends to 0.11 km above the ground, the first 0.15 km bin includes upstream PBLH values between 0.11 km and

0.26 km. By the third bin, spanning 0.41–0.56 km, the wind plant impact is no longer observable (Fig. 16(a)). Figure 16(b) separates the dataset by upstream PBLH $\leq 0.25$ km and $> 0.25$ km. As in Fig. 15(b), the bulk of the wind plant impact is contained within the $\leq 0.25$ km bin. From the lidars, the wind plant causes a 141 % increase for lower upstream PBLH and a slight decrease of 2 % for higher upstream PBLH.

Dependence on upstream PBLH is attributed to the mechanism with which wind plants entrain energy from their surround-

ings. In large wind plants, vertical kinetic energy fluxes bring in energy from above to replenish the energy extracted by the turbines (Calaf et al., 2010; Cal et al., 2010). This process generates an internal wind plant boundary layer where enhanced turbine-induced turbulent mixing occurs (Frandsen, 1992; Frandsen et al., 2006), which interacts with the planetary boundary layer. Based on the current analysis and that of Krishnamurthy et al. (2024), this internal boundary layer is expected to extend





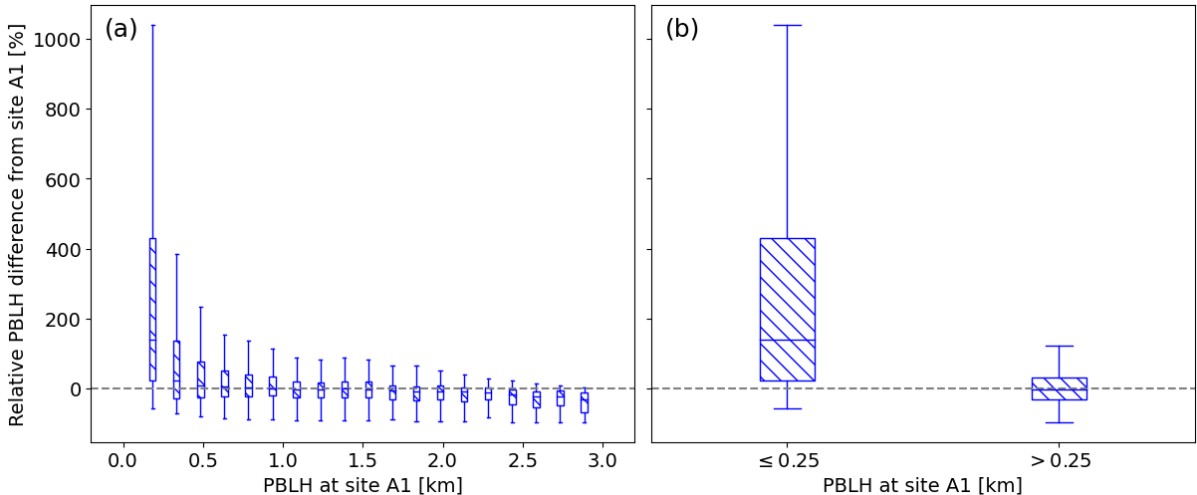

**Figure 16.** Box and whisker plots of the percent difference between PBLH measured by the lidars at the downstream sites (H) and the upstream site (A1) when wind is from the south, normalized by the upstream PBLH. The distributions are binned by upstream PBLH (a) with bin widths of 0.15 km and (b) separated into values above and below 0.25 km.

up to 0.25 km at the KP and AF wind plants, or about 0.1 km above the top tip of the turbine rotors. When PBLH is above
this height, the internal boundary layer stays within the planetary boundary layer, where ambient turbulence already exists, so PBLH is unaffected. However, when upstream PBLH is below this height, the internal boundary layer deflects the planetary boundary layer upwards as flow moves through the wind plant, increasing PBLH downstream.

## 4.5 Downstream persistence of wind plant impacts

As discussed in Sect. 1, the downstream persistence of wind plant-induced PBLH impacts is not yet well characterized. To
address this question, the PBLH distribution measured by the IR spectrometer at the ARM C1 site is compared to those measured by the three IR spectrometers nearest to KP and AF under stable, southerly wind conditions (Fig. 17(a)). When the wind is from the south, the ARM C1 site is ∼20 km downstream of the northernmost row of KP turbines (Fig. 1). One row of turbines from the Thunder Ranch wind plant lies between the ARM C1 site and the outlet of KP, but as with site G under northerly wind conditions (Sect. 4.1), this single row is not expected to significantly influence PBLH measurements.
Figure 17(a) shows that, 20 km downstream of the wind plant, PBLH has returned to a distribution that closely matches the upstream distribution, indicating that PBLH is no longer affected by the wind plant. When the ARM C1 IR spectrometer is included, 234 data points satisfy all conditional sampling criteria. The Kolmogorov–Smirnov test is performed on all pairs of sites, showing statistically significant differences between the PBLH distributions measured inside and outside of the wind plants' influence (Table 4).

To provide further insight into downstream PBLH recovery, the IR spectrometer at site E36 is used under stable, northerly wind conditions. When the wind is from the north, site E36 is ∼13 km downstream of the southernmost turbine in AF. Due




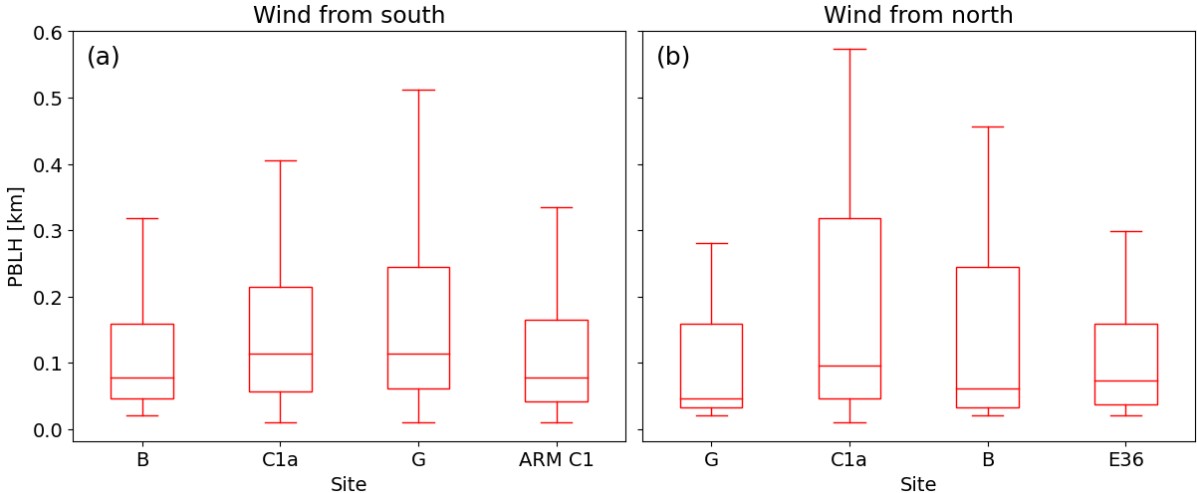

**Figure 17.** Box and whisker plots of PBLH measured by the IR spectrometers, including the ones located far downstream (see Table 2), under stable conditions, when wind is from (a) the south, with $N = 234$, and (b) the north, with $N = 81$ for sites B, C1a, and G, and $N = 54$ for site E36. Sites on the horizontal axis are arranged from upstream to downstream for each wind direction.

**Table 4.** Results of the Komogorov–Smirnov test comparing the PBLH distributions presented in Fig. 17(a), with $N = 234$, where $p < 0.05$ (in bold text) indicates a statistically significant difference between the compared samples.

| Site | C1a | G | ARM C1 |
|------|------|------|------|
| B | $\mathbf{p < 0.05}$ | $\mathbf{p < 0.05}$ | $p = 0.06$ |
| C1a | | $p = 0.96$ | $\mathbf{p < 0.05}$ |
| G | | | $\mathbf{p < 0.05}$ |

to limited data availability for northerly winds, the criterion that all data points in the distributions are concurrent in time is removed for the E36 dataset, leaving 81 data points from the the three IR spectrometers nearest to KP and AF and 54 points from the one at site E36. PBLH distributions are presented in Fig. 17(b). Under northerly wind conditions and at 13 km

downstream, the upper whisker and upper bound of the box have returned nearly to the upstream heights observed at site G. However, the median of the distribution remains elevated relative to the upstream. From these distributions, it appears that PBLH has begun to recover at 13 km downstream, but may not be fully recovered at this distance. The limited number of data points do not allow for Kolmogorov–Smirnov analysis. Comparing the complete recovery observed at 20 km downstream in Fig. 17(a) to the partial recovery at 13 km downstream in Fig. 17(b) provides insight into the downstream extent of wind plant

influence on PBLH.





# 5    Conclusions

The current study capitalizes on the extensive experimental dataset obtained by the AWAKEN campaign to investigate the impact of wind plants on PBLH, using both thermodynamic and turbulence-based methods. Three thermodynamic methods are tested by comparing their consistency across radiosonde and IR spectrometer measurements, and the Heffter method is selected

for the analysis. Scanning lidar profiles of vertical velocity variance are used to obtain the turbulence-based measurement of PBLH. Both the selected thermodynamic method and the turbulence-based method are used to compare PBLH upstream and downstream of a wind plant under different atmospheric conditions. The wind plant increases PBLH under stable conditions when flow is from the south and north, and no impact is observed under convective conditions. Binning data points by hour of the day shows the strongest impact occurring during the early hours of the night, suggesting the wind plant delays the evening

transition from convective to stable boundary layer. Wind plant impacts are shown to be most dependent on upstream PBLH. When upstream PBLH $\leq 0.25$ km, median wind plant-induced changes in PBLH are observed to be greater than 30 % using thermodynamic methods, and more than 140 % using turbulence-based methods. At 13 km downstream of the wind plant, PBLH has mostly returned to its upstream distribution. By 20 km downstream, the effects of the wind plant are no longer visible and PBLH has completely recovered its upstream distribution.

These findings provide strong evidence that wind plants can modify the planetary boundary layer in their surrounding area, though these impacts are confined to a relatively small region around the wind plant. The qualitative agreement between the thermodynamic and turbulence-based methodologies highlights the wind plant impact despite the ambiguity of the PBLH definition, particularly under stable atmospheric conditions. Future work will include further investigation into the differences between the two methods. By characterizing the turbulence and temperature profiles of the boundary layer directly, differences

in PBLH magnitude obtained using the two methods under stable conditions can be better understood. Furthermore, differences in PBLH during morning and evening transitions, as defined by the two methods, require further investigation. Finally, the current study shows partial recovery 13 km downstream of the wind plant and total recovery 20 km downstream, but does not fully characterize the recovery process or identify the factors that influence this recovery. Future investigations will use numerical tools to explore downstream persistence of PBLH impacts in more detail.

Despite these limitations, this study presents, to the best of our knowledge, the first experimental investigation of PBLH changes caused by a utility-scale wind plants that uses such a vast and diverse fleet of remote sensing instruments. The findings of the current study are invaluable for a variety of research areas. First, they can be used for validating and improving atmospheric models, as PBLH represents a key parameter that is the synthesis of a complex interplay of different phenomena that are challenging to capture. PBLH is also a major driver of wind plant performance and wake evolution. Second, this

study can inform future field campaign development by highlighting the importance of the synergy between multiple different instruments for measuring atmospheric flow over large areas. Finally, this study corroborates the hypothesis that wind plants interact with the atmosphere to affect local climatology. These findings can inform the siting of future wind plants to better understand local environmental impacts, enabling a shift toward a more multidisciplinary approach to wind energy research and deployment.



*Data availability.* All datasets are publicly available on the ARM Discovery site (https://adc.arm.gov/discovery/#/) or on the Wind Data Hub (https://wdh.energy.gov).

*Author contributions.* AA, MPu, AJ, and EM contributed to conceptualization, methodology development, and formal analysis. NH, SL, PMK, JG, JDJ, RK, RKN, and MPe were responsible for data collection and curation for each of the instruments. NB, NH, PMK, ES, and SW supervised the work. PM acquired funding and leads the AWAKEN project. AA wrote the original draft, which all authors reviewed and
edited.

*Competing interests.* The authors declare no competing interests.

*Acknowledgements.* This work was authored in part by the National Renewable Energy Laboratory, operated by Alliance for Sustainable Energy, LLC, for the U.S. Department of Energy under Contract No. DE-AC36-08GO28308. Additional work was performed under the auspices of the U.S. Department of Energy by Lawrence Livermore National Laboratory under Contract DE-AC52-07NA27344. Pacific
Northwest National Laboratory is operated for DOE by the Battelle Memorial Institute under Contract DE-AC05-76RLO1830. Funding provided by the U.S. Department of Energy Office of Energy Efficiency and Renewable Energy Wind Energy Technologies Office. This paper describes objective technical results and analysis. Any subjective views or opinions that might be expressed in the paper do not necessarily represent the views of the U.S. Department of Energy or the U.S. Government. The U.S. Government retains and the publisher, by accepting the article for publication, acknowledges that the U.S. Government retains a nonexclusive, paid-up, irrevocable, worldwide
license to publish or reproduce the published form of this work, or allow others to do so, for U.S. Government purposes. The authors would like to thank the ARM infrastructure for making the data publicly available and to all the ARM mentors for maintaining and processing the instrument data. The authors would also like to thank the Wind Data Hub team for making the AWAKEN field campaign data publicly available.





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
