# Peer review of "Operational wind plants increase planetary boundary layer height: An observational study"

_Wind Energy Science, 2024_

## Author Response (AR1)

**Reply to Reviewer 1**

**Reviewer's general comments:** *The manuscript presents new measurements from the AWAKEN campaign. This community campaign is crucial for the wind energy community and is unique in its measurement capabilities. This is a continuation of the Krishnamurthy et al. (2024) work, where additional technical details of the experiments are described. The present work reports measurements of the planetary boundary layer height at the AWAKEN site from different wind directions. The measurements reveal that wind farms increase this boundary layer height.*

**Reply:** The authors would like to thank the reviewer for their effort in reviewing our paper, and for their constructive comments and insightful recommendations. We have provided responses to each specific comment from the reviewer and made corresponding changes to our revised manuscript as presented below.

1. **Reviewer comment:** *Line 6: "Then, using one of these methods that is thermodynamic and" --> Please mention the used method and why this method has been selected (in abstract and conclusion)*

   **Reply:** We have added the following sentences to the abstract: "The Heffter method is selected as the thermodynamic method because it generates the most consistent results for the radiosonde and infrared spectrometer. A minimum vertical velocity variance method is used for a turbulence-based definition." Both methods were already mentioned in the conclusion, but some justification has been added: "…the Heffter method is selected for the analysis as it produces the most consistent estimates between the two instruments."

2. **Reviewer comment:** *Line 193: Please clarify this statement.*

   **Reply:** This sentence now reads: "For the current investigation, the value closest to the ground is always used."

3. **Reviewer comment:** *Line 234: How is the "2K" selected? Are the results sensitive to this value?*

   **Reply:** While the critical lapse rate used for the Heffter method has been shown to depend on the measurement site (Delle Monache et al. 2004), the 2 K temperature threshold has been used consistently since the method was first proposed (Marsik et al. 1995, Delle Monache et al. 2004, Snyder and Strawbridge 2004, Sivaraman et al. 2013). This temperature threshold is used to account for mixing which overshoots the base of the inversion layer (Marsik et al. 1995).

4. **Reviewer comment:** *Line 235: "If no inversion layer meeting these criteria is identified." --> How often does this happen?*

   **Reply:** This happens very infrequently. In the current study, only one profile was measured in which an inversion layer was not detected. The following statement has been added to the revised manuscript: "If no inversion layer meeting these criteria is identified, PBLH is defined

as the height of the maximum potential temperature gradient (as in the potential temperature gradient method described above), a very uncommon occurrence (<0.01%) in the current investigation."

5. **Reviewer comment:** *Section 3.2: Using the minimum in w^prime w^prime. Looking at the corresponding Figure 6, we could also consider the intersection between the behavior between 600 and 1000 meters (approximate constant) and the linear behavior between 300 and 600 meters. This would give a value closer to 600 meters compared to the present value closer to 800 meters. Different vertical bin sizes could also be used. The point is, would it be possible to give some indication of uncertainty in the obtained heights?*

   **Reply:** The bin averaging is utilized to ensure the robustness of the PBLH estimates, i.e., to ensure that the global minimum of the $w'w'$ profile is indeed detected as opposed to any local minima in the variance profile. To address the reviewer's question, we quantified the sensitivity of the PBLH estimates on the bin size for the period May 1 to October 31, 2023. Six widths (20m, 50m, 70m, 100m, 150m, 200m), including the one reported in the manuscript (100m), have been used to average the $w'w'$ profiles into bins and subsequently quantify the PBLH. The results, reported in the figure below as diurnal average, reveal high sensitivity for the 20 m and 50 m bin size during stable conditions (i.e. 00:00 to 12:00 UTC). When these bin sizes are adopted, the minimum in the bin averaged $w'w'$ profiles are too sensitive to the presence of small-scale fluctuations in the original $w'w'$ profile, which occur more frequently farther away from the ground due to the larger presence of noise in the data, which explains why the 20 m and 50 m diurnal distributions show higher values. By contrast, little differences in the diurnal averaged profiles are found for bin widths greater than or equal to 70 m, thus confirming that the current choice of bin width of 100 m does not excessively influence the daily averaged PBLH estimates.

[Figure]

6. **Reviewer comment:** *Section 4.2 vs. Section 4.3: Is this essentially a different representation of the same data?*

   **Reply:** Yes, these two sections describe the same data, but Section 4.3 divides the data into smaller bins. While Section 4.2 only separates the data into stable and convective conditions, Section 4.3 uses hourly bins to show the evolution of PBLH over the diurnal cycle. To clarify this point, we have added the following statement to the beginning of Section 4.3: "Here, we use the same data to explore further hourly variations that are missed when categorizing by stability alone."

7. **Reviewer comment:** *Section 4.2 and Section 4.3: Can you document the different p values in a table to provide a better overview for the reader?*

   **Reply:** A table showing all $p$-values for southerly wind conditions presented in Section 4.2 has been added to the revised manuscript (Table 4). Northerly winds are not included in this table because the IR spectrometers do not record enough datapoints under northerly winds for statistically significant comparison.

8. **Reviewer comment:** *Section 4.4 "Effects of upstream PBLH on wind plant impacts" --> In sections 4.2 and 4.3, the development of the PBLH is also discussed.*

   **Reply:** The reviewer is correct that the effects of upstream PBLH are incorporated into the analysis presented in Sections 4.2 and 4.3 of the original manuscript. In these previous sections, the impact of upstream PBLH is inferred based on other variables such as time of day and atmospheric stability. However, Section 4.4 of the original manuscript (4.5 in the revised manuscript) explores this relationship more explicitly by binning the data based on upstream PBLH values directly. This section sheds light on the physics governing wind plant-atmosphere interactions, providing insights into the vertical extent of the impact of wind plants on the boundary layer.

9. **Reviewer comment:** *Line 447: is it PBLH or the stability of the boundary layer (title section 4.2)? Of course, both are correlated.*

   **Reply:** As described in the previous reply, PBLH and stability are interconnected. The results presented in Sections 4.2 and 4.4 of the revised manuscript motivate the investigation in Section 4.5. In Section 4.4, the changes in the impact of the wind plants throughout the day are explored, along with the corresponding diurnal PBLH changes. It is this analysis that motivates the need for direct study of the effect of upstream PBLH.

10. **Reviewer comment:** *Line 463: Can you discuss the implications of the lidar blind spot for these measurements in more detail?*

    **Reply:** We thank the reviewer for drawing attention to this point. Because of the lidar blind zone, the impact of the wind plants on PBLH for the lowest PBLH values cannot be assessed. However, the lowest measured values still show a large impact, as demonstrated in Figure 17

of the revised manuscript. This explanation has been added to Section 4.5 of the revised manuscript: "Because of the blind zone of the lidar, PBLH values below 0.11 km cannot be detected, so the impact of the wind plant on the shallowest boundary layers cannot be evaluated. However, for the lowest values of measured upstream PBLH (the first 0.15 km bin includes values between 0.11 km and 0.26 km), the median increase in downstream PBLH is larger than 100%." Description of the blind zone has also been added to Section 2.4: "Both lidars have a blind zone, so measurements begin 110 m above the ground."

11. **Reviewer comment:** *Line 510-520: How do the mentioned downstream distances compare to the wind farm size.*

    **Reply:** In the case of the AWAKEN domain, the wind farm size is difficult to characterize due to the close proximity of Armadillo Flats (AF) and King Plains (KP), as well as their irregular shapes. However, the maximum extent of the wind farms in the north-south direction, i.e., the distance between the southernmost turbine of AF and the northernmost turbine of KP, is about 22 km. In Section 4.6 of the revised manuscript, this number is included for comparison: "For reference, the maximum north-south extent of the entire cluster of turbines within the region of interest (magenta ellipse in Fig. 1), from the southernmost turbine of AF to the northernmost turbine of KP, is 22 km." It has also been added to the Conclusion: "Note that the maximum extent of the wind plant cluster in the north-south direction is 22 km." The distances are not normalized, as is the standard practice for wind farm cluster analysis (e.g., Van Der Laan et al. 2023), because the appropriate quantity to use for normalization has not been established.

**Additional References:**

Marsik, F., Fischer, K. W., McDonald, T. D., and Samson, P. J., Comparison of Methods for Estimating Mixing Height Used during the 1992 Atlanta Field Intensive, J. Appl. Meteorol., 34, 1802–14, 1995.

Snyder, B.J. and Strawbridge, K.B.: Meteorological Analysis of the Pacific 2001 Air Quality Field Study, Atmos. Environ., 38, 5733–43, 2004.

Van Der Laan, M. P., García-Santiago, O., Kelly, M., Meyer Forsting, A., Dubreuil-Boisclair, C., Sponheim Seim, K., Imberger, M., Peña, A., Sørensen, N. N., and Réthoré, P.-E.: A New RANS-Based Wind Farm Parameterization and Inflow Model for Wind Farm Cluster Modeling, Wind Energy Sci., 8, 819–48, 2023.

**Reply to Reviewer 2**

**Reviewer's general comments:** *The manuscript presents an attempt to investigate the effect of operating wind power plants on the development of the planetary boundary layer height (PBLH) based on a network of advanced atmospheric remote sensing instrumentation during the AWAKEN field campaign. The manuscript is in general well written and structured, the applied methods are adequate for the investigations and properly described. Figures and tables are nicely formatted and relatively easy to read. The topic of a potential influence of wind power plants on PBLH fits well in the scope of WES. Nevertheless, I have two comments and concerns that will require some major revisions of the presented manuscript.*

**Reply:** The authors would like to thank the reviewer for their effort in reviewing our paper, and for their constructive comments and insightful recommendations. We have provided responses to each specific comment from the reviewer and made corresponding changes to our revised manuscript as presented below.

1. **Reviewer comment:** *Figure 2: although it is very nice to see that example, I really miss a statistical proof in form of profiles of average bias/RMSE over all the >200 profiles you assume to be valid and utilized in your investigations; in addition I would highly recommend to separate between convective and stable situations, because the temperature/humidity remote sensing by the passive radiometers should have rather different accuracies for ground based/ground near night-time inversions (for the stable cases) and the capping inversion on top of the CBL during convective conditions. This would also better reflect the fact that the daytime, convective PBLH (in BL meteorology often identified as z_i) and the PBLH in stable conditions (often denoted as h) are caused/influenced/determined by considerably different parameters and processes.*

   **Reply:** The primary reason a detailed statistical comparison between the IR spectrometer profiles and the radiosonde profiles was not conducted is because of the limited number of coincident measurements. Not every profile retrieved from the IR spectrometer has a corresponding radiosonde measurement. As discussed in Section 3.1.4 of the original manuscript, there are only 67 pairs of profiles after the quality control procedure is applied. For this reason, a few example profiles are used to illustrate the differences between the two instruments, particularly the smoothing effect of the IR spectrometer. However, to provide the reader with a better understanding of how the two instruments compare overall, we have calculated the bias and RMSE for the potential temperature profiles, and added the following paragraph to Section 2.3 of the revised manuscript:

   "To provide a quantitative comparison between the profiles measured by the radiosonde and those obtained by the IR spectrometer, the bias and root mean square error (RMSE) on potential temperature ($\theta$) are computed. For each sounding collected by the radiosonde at site H, the nearest retrieval (in time) captured by the IR spectrometer at site G is identified. After removing periods with clouds and nonzero QC flags, as described above, and limiting the time

between the sounding and the corresponding retrieval to less than 15 minutes, 67 pairs of profiles remain (N=67). The mean bias in $\theta$ between the ground and 3 km aloft is 0.49 K, while the RMSE is 3.0 K. Note that some differences between the profiles collected by the radiosonde and the IR spectrometer are expected due to the spatial offset between the two instruments, including the fact that the IR spectrometer is stationary while the radiosonde moves with the wind. Still, the low bias provides confidence in the reliability of the IR spectrometer, while the RMSE is attributed to the smoothing effect of the retrieval."

Of these 67 pairs of profiles, 51 are captured under stable conditions, while only 16 are captured under convective conditions. The example shown in Fig. 2 of the original manuscript shows the typical daily distribution of radiosonde launches, which are more concentrated at night (note that local time is UTC-5). Since stable conditions are more relevant to the current study, this distribution works well for our purposes. The mean bias and RMSE for stable profiles only are 0.41 K and 3.4 K, respectively. For convective profiles, mean bias and RMSE are 0.73 K and 1.0 K. Due to the small sample sizes, these numbers are not included in the revised manuscript.

2. **Reviewer comment:** *The main issue I have with the manuscript is that I am not convinced that it is the effect of the wind power plant (alone) that is triggering the lift in PBLH during stable conditions. The wind farms along the investigated transects are located along/across ridges in the terrain that have (at least what I can read out of the maps) elevations above the surrounding flats/valleys that are in the same order as the wind turbine heights, consequently the terrain roughness alone also might influence the PBLH downstream. Do you have any indications that could provide additional evidence that it is really (mainly) the presence of operating wind power plants that causes the lift in PBLH and not the pure presence of elevated terrain? Have there e.g. been situations with considerable wind speed where the wind farms have been shut down for some hours/days that could show that the farms not operating won't give a corresponding increase in PBLH downstream? Or are there any fine-scale model simulations in the AWAKEN context available that could be used to shed more light on this potential issue? I am fully aware that it might not be possible to revoke my concern based on the actually available data, but in this case the issue should at least be discussed in the introduction and during the presentation of the results. And maybe the corresponding conclusions should also be formulated a bit more carefully "strong evidence that wind plants can modify….. " (line 515)*

**Reply:** We thank the reviewer for raising this point. To provide further support for the hypothesis that the wind plant is, at least partially, responsible for the observed changes in PBLH, we have explored both options suggested by the reviewer. First, we checked the turbine operation data for periods of curtailment, defined here as periods where the wind speed is greater than 5 m/s (cut-in wind speed is 3 m/s) and at least 80% of turbines are producing less than 10% of their rated power. There are only about 20 datapoints from the IR spectrometers that coincide with periods of curtailment, insufficient for statistical analysis. However, because

the lidars have higher temporal resolution (10-min vs. 30-min) and a longer period of data availability (1 year vs. 5 months), 93 lidar profiles obtained during periods of curtailment are available, under stable conditions and with wind from the south. The distributions of PBLH derived from the lidars at sites A1 and H during periods of curtailment and normal turbine operation are compared in the figure below. From this figure, particularly panel (b) which shows the change in PBLH between the two sites, it is clear that much larger changes occur when the turbines are operating. Some changes in PBLH are still observed during curtailment, suggesting that terrain also plays a role in the spatial evolution of PBLH at the AWAKEN site. Still, the median difference in PBLH between the upstream and downstream sites is 9% when turbines are curtailed (terrain-induced) and 56% when turbines are operating (wind plant and terrain-induced). In the revised manuscript, we have removed curtailed periods from the rest of the analysis.

[Figure]

We have added the above figure to the revised manuscript, along with a new section (4.3 in the revised manuscript) describing the observed differences between curtailed and operational periods, and a paragraph defining curtailment and normal operation at the end of Section 4.1.

To further investigate the role of terrain, we also used an existing set of WRF simulations, generated for the AWAKEN benchmark ([awaken-benchmark.readthedocs.io](awaken-benchmark.readthedocs.io)), that had modeled the AWAKEN domain both with and without turbines. Using the Heffter method with the potential temperature extracted from a north-south transect (flow is from the south) of the domain that passes through King Plains and Armadillo Flats, we calculated PBLH for both cases. The vertical velocity variance is not resolved at the mesoscale and not computed as a WRF output, so the turbulence-based method was not used here. To isolate the impact of the wind plants, we then took the difference in PBLH between the case with turbines and the case without. The data was extracted with hourly resolution, and we focused on the PBLH difference during the nighttime hours, from 4-10 UTC (23-5 local time). The figure below presents the average and standard deviation of the difference during this period, with the extent of the wind plants indicated by the gray shaded region. Upstream of the wind plant, there is no significant difference between the cases with and without the turbines. Within the region of the

wind plant and downstream, the difference is positive overall, consistent with the observations presented in the manuscript and supporting the hypothesis that wind plants cause an increase in PBLH. The magnitude of the difference changes within the wind plant, suggesting an interaction between wind plant layout and terrain that can be explored in a future study with additional simulations. The PBLH increase extends 6-7 km downstream (to the north) of the northernmost turbine, then PBLH in the wind plant case drops below that without turbines. This downstream behavior also warrants future investigation.

While these simulation results provide additional support for the findings described in the paper, we have decided not to include them in the revised manuscript. We wish to preserve the observational nature of the study. Furthermore, the simulation results raise many additional questions that cannot be answered within the scope of the current study. We use them to inspire ideas for future study, added to the conclusion of the revised manuscript: "Current findings show some changes in PBLH during wind plant curtailment. These changes may be induced by terrain, surface roughness, or other mesoscale effects. The interaction between these effects and wind plant impacts on PBLH warrant further study, likely using mesoscale modeling tools that can accurately represent terrain."

[Figure]

3. **Reviewer comment:** *In general: "wind plant" is for me a rather unusual expression in wind energy meteorology, I would suggest to replace it by "wind power plant" and maybe even introduce an abbreviation (WPP?) as you use this expression quite a lot.*

**Reply:** The term "wind plant" has been used more commonly in recent years within the wind energy meteorology community. Several recent publications in Wind Energy Science have employed this terminology (e.g., Simley et al. 2021, Sanchez Gomez et al. 2023, Veers et al. 2023) as a shorter way of referring to wind power plants. For this conciseness and to avoid the use of additional acronyms, we have retained the use of "wind plant" within the manuscript.

4. **Reviewer comment:** *Line 6: "multiple different instruments" sounds a bit strange/weird; maybe enough with just "multiple instruments"; if you want to specify "different" in more detail you should spend another sentence with a few wore details on it.*

   **Reply:** This phrase has been revised to read: "multiple types of instruments."

5. **Reviewer comment:** *Line 145: is it really correct that the vertical resolution is 2 km at 1 km above the ground? If so, how could this in any way be useful for BL studies, in particular in defining a reliable value of the BL height?*

   **Reply:** Here the resolution indicates the amount of smoothing in the profile. This quantity is determined by the full-width half-maximum of the rows of the A-kernel matrix, an output of the TROPoe retrieval which provides the smoothing functions. While there is significant smoothing of the temperature and humidity profiles at higher elevations (see Fig. 2 of the manuscript), changes in lapse rate are captured with sufficient precision to enable estimation of the boundary layer height using the Heffter method (see Fig. 4 of the manuscript). The PBLH values tend to be slightly underestimated due to the smoothing, but the general agreement between the radiosonde-retrieved PBLH and the spectrometer-retrieved PBLH is good. Furthermore, the current study evaluates the difference in PBLH at different sites, so a consistent low bias does not strongly affect these results. Any effect of the slope<1 in the radiosonde-IR spectrometer comparison will cause, at most, a slight underestimation of the PBLH differences presented later in the study. This slight underestimate means that the resulting impact of wind plants on PBLH is a conservative estimate. Finally, the current study primarily focuses on stable conditions where PBLH is low, within the region where gradients are faithfully captured by the IR spectrometer retrievals (see again Fig. 2 of the manuscript). To clarify the meaning of "resolution" in this context, this sentence has been modified in the revised manuscript: "The vertical resolution of the TROPoe retrievals, as defined by the full-width half-maximum of the smoothing functions, starts at 6 m near the ground surface, then decreases exponentially with elevation to 2 km at 1 km above the ground."

6. **Reviewer comment:** *Line 178: please detail how you calculated/determined this scan specific threshold.*

   **Reply:** This threshold is dynamically adapted to the specific dataset, as described by Beck & Kühn (2017). When plotting the joint probability distribution of normalized radial wind speed and normalized signal to noise ratio (SNR), a sudden jump in probability is observed. Values below this jump are removed, while values above it are retained. This jump is defined as the point where the range of normalized wind speed is 25% of its maximum range across all probabilities. See the figure below for an example. This process is implemented in the publicly available FIEXTA code base (https://github.com/nrel/fiexta). To clarify this point in the revised manuscript, the sentence describing this process has been modified to read: "All the occurrences with probability below a scan-dependent threshold, determined by the range of

wind speeds within each probability bin, are considered erroneous velocity reads and removed from the time records."

[Figure]

7.  **Reviewer comment:** *Section 3.1.: I miss some more central references on the different methods to derive PBLH, in particular prior work in comparing different methods.*

    **Reply:** In the revised manuscript, we have added a paragraph summarizing relevant previous work at the beginning of Section 3.1.

8.  **Reviewer comment:** *Line 232: I don't get the meaning of "land-locked"*

    **Reply:** Here "landlocked" is used to indicate that the ARM SGP site is not near a body of water, so the lapse rate threshold used for marine boundary layers is not appropriate there.

9.  **Reviewer comment:** *Figure 4: as suggested before, I would plot this separately for the stable and convective situations.*

    **Reply:** The comparison between PBLH calculated from the radiosonde and the IR spectrometer, separated into stable and convective conditions, is shown below. With such few datapoints (N=51 for stable and N=16 for convective), calculating a least-squares regression is not very valuable. However, the coefficient of determination ($R^2$) is computed for each case. Under stable conditions, these $R^2$ values are relatively low for all three methods due to the small range of PBLH values. Still, the Heffter method shows the best agreement. Under convective conditions, the modified parcel performs slightly better than the Heffter method. However, since the phenomenon of interest in the current study occurs most prominently during stable conditions, the good performance of the Heffter method under stable conditions is weighted more heavily.

[Figure]

10. **Reviewer comment:** *Lines 273/274: you should also mention buoyancy production here in this context.*

**Reply:** We thank the reviewer for bringing this omission to our attention. We have added the following sentence to the revised manuscript: "During sunny days, buoyancy lifts parcels of air from the surface, enhancing the vertical component of turbulence."

11. **Reviewer comment:** *References: Here there are numerous inconsistencies with respect to abbreviating journal names, the missing of upper case in journal names, the presentation of doi, and some reference incompleteness*

- *Acevedo (line 557): journal name lower case*

- *Baars (line 563): remove "Publication Title:"; journal name abbreviated*

- *Beck (line 568): journal name lower case*

- *Bodini (line 74): journal name abbreviated*

- *Cal (line 580): journal name lower case*

- *Dai (line 590): journal name abbreviated*

- *Delle (line 596): journal name abbreviated*

- *Duncan (line 600): journal name abbreviated*

- *Frandsen (line 612): remove "An International Journal for ……."*

- *Hauke (line 623): journal name lower case*

- *Jozef (line 635): journal name abbreviated*

- *Krishnamurthy (line 645): journal name missing*

- *Lee (line 647): journal name abbreviated*

- *Li (line 663): journal name abbreviated*

- *Lu (line 671): journal name abbreviated*

- *Münkel (line 697): journal name lower case*

- *Neggers (line 604): journal name abbreviated*

- *Pichugina (line 713): journal name lower case*

- *Su (line 737): journal name abbreviated*

- *Tucker (line 740): journal name lower case*

- *Turner (line 743): journal name abbreviated*

- *Wu (line 760): journal name lower case*

**Reply:** We have corrected the references in the revised manuscript to be consistent with each other and with the requirements of the journal.

**Additional References:**

Simley, E., Fleming, P., Girard, N., Alloin, L., Godefroy, E., and Duc, T.: Results from a wake-steering experiment at a commercial wind plant: investigating the wind speed dependence of wake-steering performance, Wind Energy Sci., 6, 1427–1453, 2021.

Sanchez Gomez, M., Lundquist, J. K., Mirocha, J. D., and Arthur, R. S.: Investigating the physical mechanisms that modify wind plant blockage in stable boundary layers, Wind Energy Sci., 8, 1049-1069, 2023.

Veers, P. et al.: Grand challenges in the design, manufacture, and operation of future wind turbine systems, Wind Energy Sci., 8, 1071–1131, 2023.

**Reply to Reviewer 3**

**Reviewer's general comments:** *The study focuses on whether wind turbine operation modifies boundary layer heights downstream, as observed in the AWAKEN field campaign. It uses and compares multiple methods of determining PBL heights from multiple instruments.*

*The manuscript is well-structured, with clearly described methods suitable for the investigations. The figures and tables are well-formatted and easy to read. The impact of wind power plants on PBLH aligns with WES's scope. However, I am concerned about better substantiating the manuscript's conclusions.*

**Reply:** The authors would like to thank the reviewer for their effort in reviewing our paper, and for their constructive comments and insightful recommendations. We have provided responses to each specific comment from the reviewer and made corresponding changes to our revised manuscript as presented below.

1. **Reviewer comment:** *By the end of filtering out all the available data, only a few samples remain, mainly when stability constraints are applied. I am satisfied with the statistical testing used to compare the means of upstream and downstream PBLH. However, I wonder whether other differences in terrain or mesoscale circulations could be causing the changes in PBLH, which are unrelated to the operation of the wind farms. There are two ways around this. (1) Find periods when the turbines are not operating and verify that the PBL is unchanged in those cases, or (2) find some NWP model output and verify that similar changes in PBLH are not seen in the base (no wind farm) simulations but do appear when the wind farms are included. I don't suggest that you do these runs, but many groups have already run simulations with and without wind farms for the AWAKEN region. The data might already be available in the DOE repository. Of course, (1) will be best, but it might be hard to find enough data. (2) is not proof, but it might substantiate your hypothesis.*

   **Reply:** We thank the reviewer for raising this point. To provide further support for the hypothesis that the wind plant is, at least partially, responsible for the observed changes in PBLH, we have explored both options suggested by the reviewer. First, we checked the turbine operation data for periods of curtailment, defined here as periods where the wind speed is greater than 5 m/s (cut-in wind speed is 3 m/s) and at least 80% of turbines are producing less than 10% of their rated power. There are only about 20 datapoints from the IR spectrometers that coincide with periods of curtailment, insufficient for statistical analysis. However, because the lidars have higher temporal resolution (10-min vs. 30-min) and a longer period of data availability (1 year vs. 5 months), 93 lidar profiles obtained during periods of curtailment are available, under stable conditions and with wind from the south. The distributions of PBLH derived from the lidars at sites A1 and H during periods of curtailment and normal turbine operation are compared in the figure below. From this figure, particularly panel (b) which shows the change in PBLH between the two sites, it is clear that much larger changes occur when the turbines are operating. Some changes in PBLH are still observed during curtailment,

suggesting that terrain also plays a role in the spatial evolution of PBLH at the AWAKEN site. Still, the median difference in PBLH between the upstream and downstream sites is 9% when turbines are curtailed (terrain-induced) and 56% when turbines are operating (wind plant and terrain-induced). In the revised manuscript, we have removed curtailed periods from the rest of the analysis.

[Figure]

We have added the above figure to the revised manuscript, along with a new section (4.3 in the revised manuscript) describing the observed differences between curtailed and operational periods, and a paragraph defining curtailment and normal operation at the end of Section 4.1.

To further investigate the role of terrain, we also used an existing set of WRF simulations, generated for the AWAKEN benchmark (awaken-benchmark.readthedocs.io), that had modeled the AWAKEN domain both with and without turbines. Using the Heffter method with the potential temperature extracted from a north-south transect (flow is from the south) of the domain that passes through King Plains and Armadillo Flats, we calculated PBLH for both cases. The vertical velocity variance is not resolved at the mesoscale and not computed as a WRF output, so the turbulence-based method was not used here. To isolate the impact of the wind plants, we then took the difference in PBLH between the case with turbines and the case without. The data was extracted with hourly resolution, and we focused on the PBLH difference during the nighttime hours, from 4-10 UTC (23-5 local time). The figure below presents the average and standard deviation of the difference during this period, with the extent of the wind plants indicated by the gray shaded region. Upstream of the wind plant, there is no significant difference between the cases with and without the turbines. Within the region of the wind plant and downstream, the difference is positive overall, consistent with the observations presented in the manuscript and supporting the hypothesis that wind plants cause an increase in PBLH. The magnitude of the difference changes within the wind plant, suggesting an interaction between wind plant layout and terrain that can be explored in a future study with additional simulations. The PBLH increase extends 6-7 km downstream (to the north) of the northernmost turbine, then PBLH in the wind plant case drops below that without turbines. This downstream behavior also warrants future investigation.

While these simulation results provide additional support for the findings described in the paper, we have decided not to include them in the revised manuscript. We wish to preserve the observational nature of the study. Furthermore, the simulation results raise many additional questions that cannot be answered within the scope of the current study. We use them to inspire ideas for future study, added to the conclusion of the revised manuscript: "Current findings show some changes in PBLH during wind plant curtailment. These changes may be induced by terrain, surface roughness, or other mesoscale effects. The interaction between these effects and wind plant impacts on PBLH warrant further study, likely using mesoscale modeling tools that can accurately represent terrain."

[Figure]

**Reviewer comment:** *I find it strange to call wind farms "wind plants". I would suggest "wind power plants" or "wind farms". But this is just what I am used to.*

**Reply:** The term "wind plant" has been used more commonly in recent years within the wind energy meteorology community. Several recent publications in Wind Energy Science have employed this terminology (e.g., Simley et al. 2021, Sanchez Gomez et al. 2023, Veers et al. 2023) as a shorter way of referring to wind power plants. "Wind plant" also emphasizes the power generating capabilities of a collection of turbines over the less technical "wind farm". For these reasons, we have retained the use of "wind plant" within the manuscript.

2. **Reviewer comment:** *L10-11 "At a site 20 km downstream of the wind plant, these effects are no longer observed, suggesting PBLH has recovered." It is a weird sentence. The PBL has recovered from what?*

**Reply:** This sentence has been modified in the revised manuscript for clarity: "At a site 20 km downstream of the wind plant, these effects are no longer observed, suggesting PBLH is not influenced by the wind plant at this distance."

3. **Reviewer comment:** *The last sentence of the abstract is very vague: "The results of this investigation show that wind plants can modify the surrounding atmosphere, improving understanding of wind plant–atmosphere interaction that is crucial for model development and validation." This is true for any study. Please be more specific.*

   **Reply:** We have revised this sentence in the manuscript: "The results of this investigation show that wind plants can modify PBLH in their vicinity. As PBLH is a key parameter for numerical models, this insight into wind plant-atmosphere interactions can be used to validate and improve the accuracy of wind energy forecasts."

4. **Reviewer comment:** *L24-25: "Furthermore, wind plant wake effects are not vertically confined to the rotor area and can extend well into the planetary boundary layer, " wake effects are in the PBL, so the sentence does not make sense. Please rewrite.*

   **Reply:** This sentence has been revised to read: "Furthermore, wind plant wake effects can extend far above the top of the turbine rotor, even impacting the planetary boundary layer height."

5. **Reviewer comment:** *L58-59: "The wind plant also lifted the nose height of the nocturnal low-level jet". Again, the wind farms don't lift anything. Please rewrite.*

   **Reply:** This sentence has been revised to read: "The wind plant also caused the nose height of the nocturnal low-level jet (LLJ) to increase by ~250 m…"

6. **Reviewer comment:** *L66. "... used WRF with the Fitch..." to "... used the WRF model with the Fitch..."*

   **Reply:** This sentence has been revised as suggested.

7. **Reviewer comment:** *Section 4.1. I am not sure you wrote at what height is the Obukhov length computed. Also, I would suggest that "The Obukhov length, L, is used to determine atmospheric stability. " is replaced by "The Obukhov length, L, is used to stratify the data by stability conditions."*

   **Reply:** We thank the reviewer for pointing out this omission. The Obukhov length is computed at 4 m above the ground. We have added this information to the description of the surface flux stations in Section 4.1. We have also revised the sentence above as suggested.

8. **Reviewer comment:** *In Figures 9, 10, 11, 12, and 17, it isn't easy to see how the distributions have changed. Please add some horizontal lines to make it easier.*

   **Reply:** We have added horizontal lines to these figures to make the plots easier to interpret.

9. **Reviewer's comment:** *The equation in L414 is very "ugly". Words are enough, and the equation can be removed.*

   **Reply:** We have removed this equation as suggested.

**Additional References:**

Simley, E., Fleming, P., Girard, N., Alloin, L., Godefroy, E., and Duc, T.: Results from a wake-steering experiment at a commercial wind plant: investigating the wind speed dependence of wake-steering performance, Wind Energy Sci., 6, 1427–1453, 2021.

Sanchez Gomez, M., Lundquist, J. K., Mirocha, J. D., and Arthur, R. S.: Investigating the physical mechanisms that modify wind plant blockage in stable boundary layers, Wind Energy Sci., 8, 1049-1069, 2023.

Veers, P. et al.: Grand challenges in the design, manufacture, and operation of future wind turbine systems, Wind Energy Sci., 8, 1071–1131, 2023.